# Mutational dissection of a hole hopping route in a lytic polysaccharide monooxygenase (LPMO)

Iván Ayuso-Fernández [1] ✉, Tom Z. Emrich-Mills [1], Julia Haak[2,3], Ole Golten [1], Kelsi R. Hall [1], Lorenz Schwaiger [4], Trond S. Moe[1], Anton A. Stepnov[1], Roland Ludwig [4], George E. Cutsail III [2,3], Morten Sørlie [1], Åsmund Kjendseth Røhr [1] & Vincent G. H. Eijsink [1] ✉

Oxidoreductases have evolved tyrosine/tryptophan pathways that channel highly oxidizing holes away from the active site to avoid damage. Here we dissect such a pathway in a bacterial LPMO, member of a widespread family of C-H bond activating enzymes with outstanding industrial potential. We show that a strictly conserved tryptophan is critical for radical formation and hole transference and that holes traverse the protein to reach a tyrosine-histidine pair in the protein's surface. Real-time monitoring of radical formation reveals a clear correlation between the efficiency of hole transference and enzyme performance under oxidative stress. Residues involved in this pathway vary considerably between natural LPMOs, which could reflect adaptation to different ecological niches. Importantly, we show that enzyme activity is increased in a variant with slower radical transference, providing experimental evidence for a previously postulated trade-off between activity and redox robustness.

Electron hopping and hole hopping are important mechanisms in biological redox reactions that serve to swiftly transfer charge between redox species[1,2]. Hopping pathways in proteins are primarily comprised of strings of aromatic and redox-active amino acid side chains. Their presence is thought to stem from an expansion of the amino acid repertoire during evolution; in an increasingly oxidizing aqueous environment, tyrosine (Tyr, Y) and tryptophan (Trp, W) in solution provided a primordial protection against oxidative damage that was later incorporated into enzymes as built-in networks[3]. Electron/hole hopping via Tyr and Trp chains within proteins in organisms existing today provides protection from oxidative damage[4] but has also been shown to be implicated in an array of highly specialized functions, including direct oxidation of recalcitrant lignin in heme peroxidases[5],

DNA repair in photolyases[6] and the fascinating cryptochromes that are part of the magnetic avian compass[7], to name just a few. Hole hopping is not limited to monomeric proteins, as illustrated by class I ribonucleotide reductases, where catalysis involves a hole traversing from one subunit to another through a chain of aromatic amino acids covering an impressive distance of 35 Å[8–11].

Hole hopping is thought to be initiated by a strongly oxidizing species (the initial hole) in the active site of an enzyme, often involving a metal cofactor[1]. In the absence of a substrate, these oxidizing species can react nonspecifically with surrounding functional groups leading to enzyme damage and inactivation, with a subsequent cost for the cell. To avoid this, hole hopping paths paved with the side chains of Tyr and Trp have evolved allowing for rapid transfer of holes away from

[1]Faculty of Chemistry, Biotechnology and Food Science, Norwegian University of Life Sciences (NMBU), 1432 Ås, Norway. [2]Max Planck Institute for Chemical Energy Conversion, Stiftstrasse 34-36, 45470 Mülheim an der Ruhr, Germany. [3]Institute of Inorganic Chemistry, University of Duisburg-Essen, 45141 Essen, Germany. [4]Biocatalysis and Biosensing Laboratory, Department of Food Sciences and Technology, Institute of Food Science and Technology, University of Natural Resources and Life Sciences (BOKU), Muthgasse 18/2, Vienna 1190, Austria. ✉e-mail: ivan.ayuso-fernandez@nmbu.no; vincent.eijsink@nmbu.no

the active site, to be defused from the protein surface by reductants[12]. Phrased differently, hole hopping entails that an electron is channelled through the protein towards the catalytic center, where it is used to neutralize a radical, for example converting a hydroxyl radical, OH· to a harmless hydroxy, OH⁻.

The fundamental importance of this process is evidenced by an abundance of predicted hole hopping chains in known oxidoreductases: ~40% of all oxidoreductases include strings of three or more Tyr/Trp residues within a distance that would allow for hole hopping (up to around 10 Å)[1]. While the presence and importance of hole hopping routes are generally accepted in redox biochemistry, this process has been studied in detail only for a limited number of oxidoreductases[2,13]. Investigating hole hopping mechanisms in novel enzymes and enzyme families will increase our understanding of this fundamental protective and functional mechanism, while accumulating insight may be harnessed for engineering better biocatalysts[12,14,15].

Lytic polysaccharide monooxygenases (LPMOs), discovered less than two decades ago[16,17], are a family of widely spread oxidoreductases with multiple functions and considerable industrial potential[18–23]. LPMOs are mono-copper enzymes[24,25] capable of selectively activating C-H bonds in a variety of substrates including crystalline chitin[17] and cellulose[24–26], two of the most abundant sources of renewable carbon on Earth. LPMOs are currently classified into 8 families in the carbohydrate-active enzymes database (CAZy; http://www.cazy.org/)[27] as auxiliary activity (AA) families 9–11 and 13–17. They are mainly found in fungi and bacteria but also in insects, plants and viruses. Although they have been traditionally studied in the context of biomass degradation, recent publications highlight a diversity of (potential) LPMO functions[20], related to fungal[28–30], oomycete[31] and bacterial[32–34] pathogenesis, insecticidal activity[35], mycorrhizal[36] and commensalism[37,38] symbiosis, insect development[39] and fungal cell wall remodelling[40].

While LPMOs were originally considered monooxygenases using $O_2$ as the co-substrate, hence their name, recent studies show that the favoured co-substrate is $H_2O_2$, and that LPMOs in fact are peroxygenases[41–46]. The peroxygenase reaction requires reductive activation of the enzyme (i.e., reduction of the copper) and involves the formation of a hydroxyl radical, resulting from homolytic cleavage of $H_2O_2$, and a copper-oxyl intermediate[41,43,46–49]. Like most oxidoreductases, LPMOs suffer from oxidative damage[41,50,51]. In the presence of excess $H_2O_2$ and/or absence of substrate, autocatalytic oxidation of the Cu-coordinating histidines can lead to enzyme inactivation[41,51].

Recent studies have detected the formation of amino acid radicals in fungal AA9-type LPMOs during reactions with $H_2O_2$, with a potential role in protection rather than substrate oxidation[43,52–54]. In these studies, using stopped-flow UV-Vis absorption spectroscopy, a prominent tyrosyl radical (Y·) was consistently identified, while some studies also demonstrated the presence of a tryptophanyl radical (W·)[43,52–54]. The formation of these radicals is thought to be initiated by highly oxidizing intermediates that emerge during the catalytic cycle and that can initiate amino acid radical formation, namely the hydroxyl radical formed after homolytic cleavage of $H_2O_2$[43,46,47] or the subsequently formed copper-oxyl[51,55]. Though these previous studies speculate about the involvement of Y· and W· in hole hopping protective mechanisms, a thorough, conclusive investigation of hole hopping in LPMOs, linking radical formation to LPMO functionality is lacking. We envisaged that studying these protective mechanisms in LPMOs could be particularly informative because the combination of a powerful redox reaction with an insoluble, not rapidly diffusing substrate could make these enzymes particularly vulnerable to the devastating effects of off-pathway reactions.

As outlined below, several bacterial LPMOs, belonging to the AA10 family, contain structural elements that suggest the presence of conserved hole-hopping routes that traverse the enzyme. Here, we have combined phylogenetics and sequence space analysis with extensive protein mutagenesis, detection of fast amino acid radical formation and precise quantification of LPMO activity to map the charge transfer in LPMOs and link it to reactivity on true substrates. Moreover, by manipulating the charge transfer velocity we were able to increase the peroxygenase activity of an LPMO, demonstrating a delicate balance between substrate oxidation and protective off-pathway reactions in this important family of oxidoreductases. Our findings provide insight into hole hopping and LPMO functionality, and open new avenues for understanding and engineering the performance of oxidoreductases.

## Results

### Sequence space analysis of LPMO cores

To identify hypothetical hole hopping routes in LPMOs, we searched the sequence space of all AA10 LPMOs for chains of aromatic residues present in the interior (core) of each enzyme. We used SmAA10A (also known as CBP21) as a structural template for this search. SmAA10A is an extensively characterized LPMO[16,17,47,50,56,57] with a core configuration of aromatic residues that, considering both the nature of the side chains and their spatial arrangement (Fig. 1[12,58]), could enable hole hopping. SmAA10A can be purified with ease and has proven tolerant to mutagenesis, thus potentially allowing the generation of multiple variants with mutated core residues. We initially selected six positions within the core of SmAA10A for analysis: F187, W178, W119, W108, Y121 and H160, as shown in Fig. 1. H160 was included as tyrosine residues participate in hole hopping routes when they have a suitable proton acceptor, which is frequently a His[1,59], and position 160 on the surface was the only reasonable option. These six positions, referred to hereafter as the core configuration, form a potential hole hopping path running from near the copper site (W178, F187) to the opposing surface (Y121, H160) via two tryptophan residues in the core.

Using an in-house script, we identified patterns of conservation in the AA10 family in positions analogous to those of the SmAA10A core configuration (Fig. 1, pie charts; underlying phylogeny shown in Supplementary Fig. 1). This analysis revealed differences in core configuration between the different AA10 groups (Fig. 1, pie charts). In chitin-active LPMOs (2085 sequences out of 3031 analysed), 43 % of single domain enzymes (970 sequences) and a striking 72 % of catalytic domains coupled to a CBM (1115 sequences) possess the same configuration as SmAA10A (highlighted in red in Fig. 1). Most of the alternative core configurations in chitin-active LPMOs deviate from SmAA10A by only one or two of the six residues, and much of the variation concerns replacement of Tyr by Trp or vice versa, potentially maintaining the ability to transfer holes. AA10 LPMOs active on cellulose or on both cellulose and chitin (referred to as "mixed") containing a CBM (713 sequences; Fig. 1, green chart) show a lower abundance of Tyr or Trp chains, and a higher variability of configurations. Still, 42% of the sequences in the dataset possess a conserved pair of tryptophan residues in the positions homologous to W178 and W119 of SmAA10A, with, notably, W178 being fully conserved. Single-domain, mixed activity LPMOs appear to be rare in Nature, represented by only 233 sequences in the dataset (Fig. 1, blue chart). Analysis of these 233 sequences revealed only one clearly conserved residue: a tryptophan in the position homologous to W178 in SmAA10A.

The aromatic residue closest to the copper (4 Å in SmAA10A) is a highly conserved phenylalanine (F187) that is replaced by tyrosine in a fraction of AA10 LPMOs. Interestingly, almost all AA9-type fungal LPMOs have a tyrosine in this position and mutagenesis studies have shown that this tyrosine is responsible for observed tyrosyl radical features[43]. While phenylalanine residues are not usually considered to contribute to hole hopping, Fig. 1 clearly highlights the strict conservation of another nearby aromatic amino acid, W178, in AA10 LPMOs, regardless of phylogenetic origin or modularity. The closest distance between W178 and the copper is 5.7 Å, well compatible with hole hopping[1]. Based on these observations, we hypothesized that a hole hopping route involving some or all of the core aromatic residues

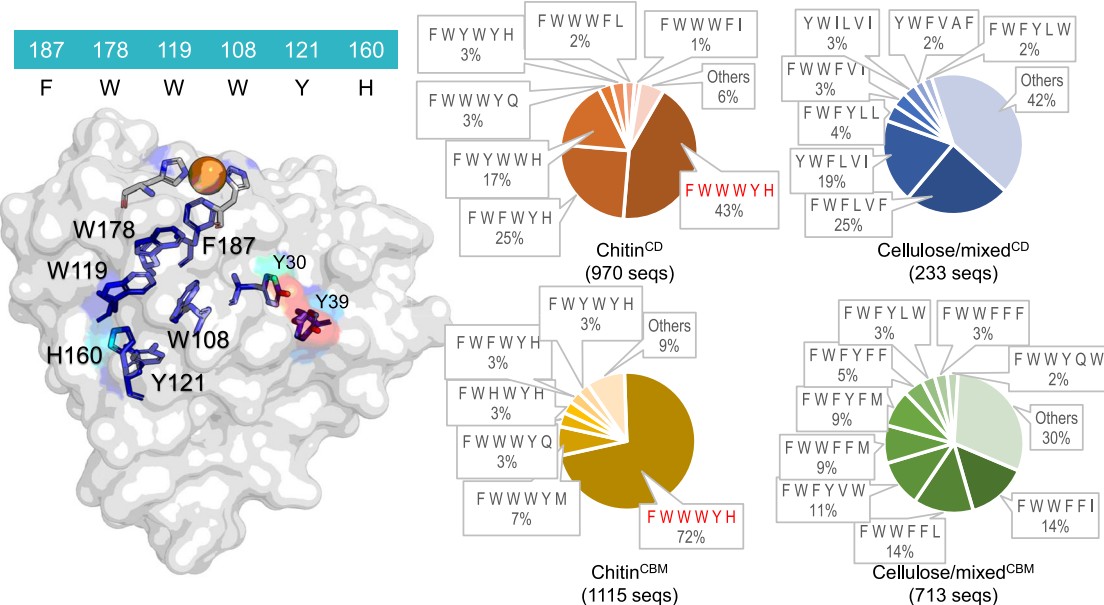

**Fig. 1 | Configurations of the cores of AA10 LPMOs.** 3031 sequences were clustered by domain modularity (CD: single, catalytic domain LPMOs; CBM: LPMOs with one or more extra carbohydrate binding modules) and substrate specificity (chitin or cellulose/mixed activity). The structure on the left shows WT *Sm*AA10A and highlights the core of six aromatic amino acids considered in this study. The colour of each amino acid indicates a scale of solvent exposure, with red being most solvent exposed and dark blue least exposed, as predicted from the PDB structure (2bem) using PyMOL. Y30 and Y39 are also shown for reasons explained in the main text. The six studied core amino acids are listed in one-letter code above the structure with residues located closest to the copper site placed to the left. The pie charts to the right show the frequency of residues at these six positions in all 3031 analyzed AA10 LPMOs as Chitin$^{CD}$ (red), Chitin$^{CBM}$ (yellow), Cellulose/mixed$^{CD}$ (blue) and Cellulose/mixed$^{CBM}$ (green). The category "Others", with 6%, 9% 42% and 30% for chitin$^{CD}$, chitin$^{CBM}$, cellulose/mixed$^{CD}$ and cellulose/mixed$^{CBM}$, comprises 26, 16, 59 and 51 different configurations, respectively.

may exist in AA10 LPMOs. Using the sequence space analysis as a guide and backed up by rational and Rosetta-based design, we generated a set of variants covering most positions of the aromatic core in *Sm*AA10A and studied mutational effects on enzyme behaviour and stability during redox stress.

## Mapping radical formation in *Sm*AA10A variants

To support the assertion of a hole hopping pathway in AA10s, we first compared the ability of wild type (WT) and mutated *Sm*AA10A to scavenge radicals formed upon reaction of the reduced Cu(I) enzyme with $H_2O_2$, using stopped-flow rapid UV-Vis absorption spectrophotometry. As shown in Fig. 2, spectra for WT *Sm*AA10A indicate efficient formation and subsequent dissipation of UV-Vis signals, with two clear features around 412 nm and 505 nm that appear and decay at approximately the same time. Based on literature values for amino acid radicals in solution[60], radicals in designed proteins[2] and similar radicals identified in LPMOs[43,52], the feature around 412 nm likely corresponds to Y·, while the second, broader feature around 505 nm corresponds to W·. In the case of WT *Sm*AA10A, both the 412 nm and 505 nm features reached their maximum absorbance values after 350 ms and decayed in 2000 ms. Based on the molar absorption coefficient values for radicals in solution from[60] (1900 mM$^{-1}$· cm$^{-1}$ for W· and 2600 mM$^{-1}$ cm$^{-1}$ for Y·) the maximum fraction of WT *Sm*AA10A containing a detectable Y· or W· is 17 % and 3 %, respectively. The rate of generation of these UV-Vis signals is independent of pH between 6.5 and 8.0, while their decay is slower at the lowest pH tested (6.5; Supplementary Fig. 2A–D). Both rates are also unaffected by the reductant used to generate the LPMO-Cu(I) (Supplementary Fig. 2E, F). The rate of generation, but not the decay rate, shows a strong dependence on the $H_2O_2$ concentration (Supplementary Fig. 3). Of note, the formation of the radical features is strongly dependent on temperature (Supplementary Fig. 2G); only at the lowest temperature tested (4 °C) radical formation and decay were slow enough to be observable.

## Mapping of the tryptophanyl feature

Mutation of W178 showed that W178F and W178M are UV-Vis silent (Fig. 2, bottom right panels), although a minute putative Y· feature became visible upon spectral subtraction (Supplementary Fig. 5). On the other hand, W178Y displayed clear differences in the behaviour of the 412 nm and 505 nm features compared with WT (Fig. 2, yellow spectra): the tyrosyl feature was smoother and broader than in the WT, with an apparent blue-shift from 412 nm to 400 nm, while the tryptophanyl signal was less distinct than in WT, though still observable. Furthermore, the evolution and decay of each signal was about one order of magnitude slower compared to the WT: maximum signals were observed after 2.5 s, while their decay took ~30 s. In W178Y, full decay to resting state was not observed for either feature during the time monitored.

Assignment of the 400 nm feature in the spectra of the W178Y variants to a tyrosyl radical is not obvious, since the energy of its electronic absorption band is high, compared to tyrosyl radicals in solution[61], or in natural[43,62] and designed proteins[2]. Spectral subtractions to remove background signals (Supplementary Fig. 5) confirmed that a blue shift indeed occurs when comparing the WT enzyme with W178Y containing variants but that this is more of a peak broadening effect rather than a true shift from 412 nm to 400 nm. The difference spectra suggest that the 412 nm feature is still present in the W178Y containing variants and also show a weak 400 nm feature in WT spectra. The feature at lower wavelength could correspond to a second feature observed usually at 15 nm less than the maximum signal for tyrosyl radicals in solution[63]. It is conceivable that the observed blue shift is due to the (non-natural) environment of the tyrosine in position 178.

The activity of the enzyme variants was tested using apparent "monooxygenase conditions", which implies that the reaction is reductant-driven and that its rate, as well as possible auto-catalytic enzyme inactivation, are determined by the level of in situ generated

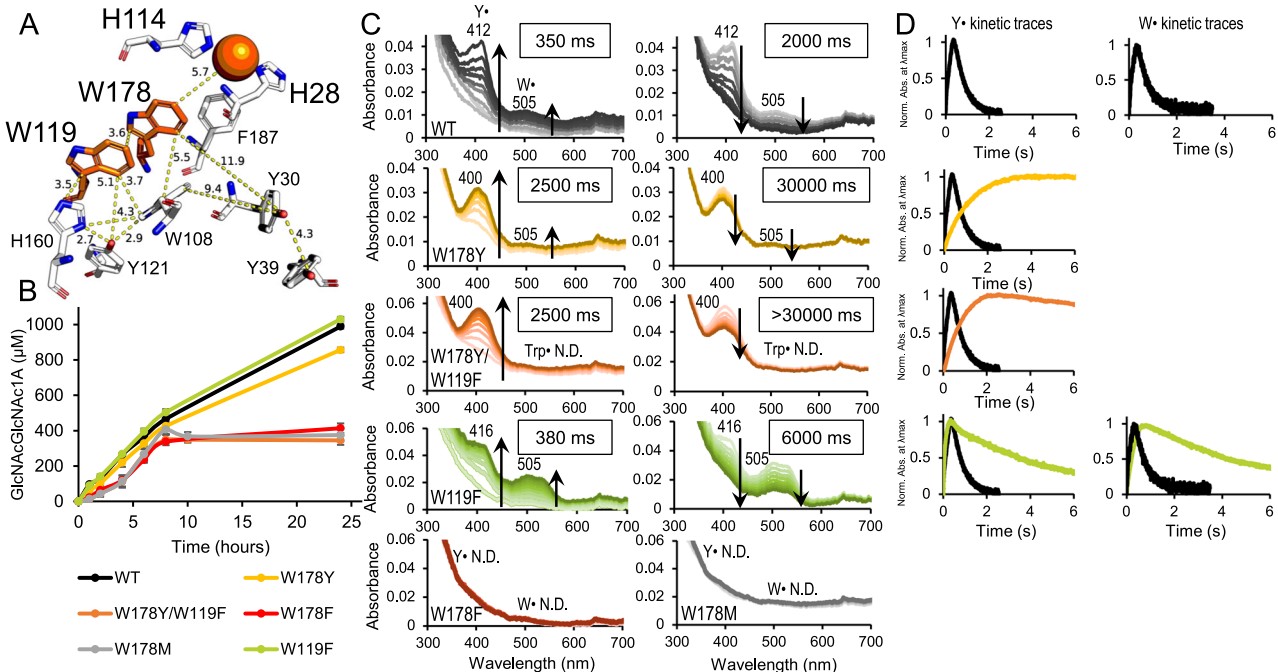

**Fig. 2 | Effects of mutating W178 and W119 on LPMO performance and radical formation and decay.** WT (black), W178Y (yellow), W178Y/W178F (orange), W119F (lime), W178F (red) and W178M (grey) are shown. **A** Cu-site, aromatic redox core and two additional tyrosine residues, Y30 and Y39, in *Sm*AA10A (pdb 2bem). **B** oxidation of chitin over time in reactions containing 1 μM LPMO, 10 g·L⁻¹ chitin and 1 mM ascorbate performed at 37 °C in 50 mM Tris, pH 8.0, represented as mean values ± s.d. (*n* = 3; see Supplementary Fig. 4A for an overview of time-courses for all enzyme variants). **C** UV-Vis spectra obtained by stopped-flow absorption spectroscopy for reactions containing 75 μM LPMO-Cu(I) and 20 molar excess H₂O₂, at 4 °C and in 50 mM Tris, pH 8.0. The features detected were assigned to tyrosyl (~412 nm) or tryptophanyl (~505 nm) radicals according to the literature[2,43,52,60]. The left panels

with spectra show formation until the maximum signal was detected and the right panels with spectra show the decay from maximum signal until resting state, with the times needed to reach maximum signals and full decay indicated in boxes. Features which could not be detected are indicated by N.D. **D** kinetic traces for the tyrosyl or tryptophanyl radicals detected in each variant. WT traces are shown in each panel for reference. Note that a weak tryptophanyl feature is visible in the spectra for W178Y and that a kinetic trace could not be derived due to the high noise. Spectral subtractions for the formation of spectral features until maximum signal are provided in Supplementary Fig. 5. Figure 6 provides a summarizing overview of formation and decay velocities for various enzyme variants.

H₂O₂[64]. These activity assays showed that the UV-Vis silent variants, W178F and W178M, are more prone to enzyme inactivation (Fig. 2, bottom left), while W178Y, still showing amino acid radical formation, was approximately as stable as the wild-type enzyme in this type of reactions. Taken together, these data point to a role for W178 in the dissipation of damaging radicals during LPMO catalysis.

Based on the sequence-space analysis and the protein structure, we hypothesised that W119 is the second residue in a possible hole hopping route through *Sm*AA10A and could thus be co-responsible for the tryptophanyl feature. W119 and W178 show a parallel displaced configuration (Fig. 2, top left) that enables the interactions needed for hole hopping[58]. Rosetta simulations identified mutation to Phe as the lowest energy solution. While replacing Trp with Phe could still allow aromatic ring coupling with W178 in theory, previous studies have shown that Phe impedes transference of holes relative to Trp[65–67]. Figure 2 shows that the W119F mutation had a clear impact on radical features. The shape of the tyrosyl feature was modified to a shoulder, while the 505 nm feature was enhanced, indicating an ~4-fold increase in the maximum fraction of enzyme containing a detectable W• relative to WT. The rates of formation of both features were similar to WT, but decay was some 3-fold slower (Fig. 2), potentially indicating that the tryptophanyl radicals were scavenged less efficiently than in WT. The relatively strong feature at 505 nm suggests that impeding hole transference through W119 promotes the accumulation of W• in position 178.

The W178Y/W119F variant (Fig. 2, orange spectra) did not show any feature at 505 nm, while showing the same slow accumulation and decay of the 400 nm feature as W178Y. We interpret these data to

mean that when transference through position 119 is blocked but there is a Tyr in position 178, the LPMO only accumulates tyrosyl radicals, in this case likely primarily Y178 radicals (see below), upon reaction with H₂O₂. It is interesting to note the apparent difference in enzyme stability under turnover conditions: while W178Y, which could still allow transference of holes from the copper site into the hydrophobic core showed similar stability as the WT, W178Y/W119F, in which further dissipation may be blocked, was more rapidly inactivated, similar to W178F and W178M (Fig. 2, bottom left panel).

Although different from the WT, W119F still showed formation and decay of a spectral feature at ~416 nm, suggesting that Y• is still generated downstream of position 119. This may mean that a phenylalanine at position 119 can still participate in transference of holes or that this position can be bypassed. One alternative path could include W108, the third tryptophan in the conserved core configuration of *Sm*AA10A. W108 has a T-shape orientation relative to W119 (Fig. 2), another common form of coupling between aromatic rings in proteins[12,58]. Unfortunately, none of the variants designed to study the role of this residue (W108A, W108F, W108Y, W108M) could be successfully expressed, indicating that position 108 might have a critical structural role. Of note, the absence of a tryptophanyl feature in the W178Y/W119F variant, which likely accumulates tyrosyl radicals in position 178, suggests that hole transfer from W/Y178 to W108 does not occur. The nature of the tyrosyl feature is discussed further below.

## Mapping of the tyrosyl feature

While fungal AA9 LPMOs contain a conserved tyrosine close to the copper that is known to form radicals[43,52,54], there are no immediately

obvious candidates for explaining the tyrosyl radical feature observed with *Sm*AA10A. In an effort to identify the species responsible for the strong Y˙-associated feature observed in *Sm*AA10A UV-Vis spectra, we targeted tyrosine residues positioned within a distance of the three core tryptophan residues that may allow hole hopping. As shown in Fig. 3 (top left), Y30, Y39 and Y121 are the only tyrosine residues within range for hole hopping, where Y121 is part of the conserved aromatic core. Y121 is positioned at 2.9 Å from W108 and 5.1 Å from W119 (distances refer to the closest contact between side chain carbon/nitrogen atoms). Y30 and Y39 are solvent-exposed, on the opposite side of the enzyme, and further away, the closest distances being 9.4 Å for Y30 and W108 and 11.9 Å for Y30 and W178. Y39 is connected to Y30 through a stacking interaction that would likely allow hole transference. Rosetta simulations suggested that replacement of Tyr by Phe was feasible at all three selected positions.

The Y30F and Y39F variants showed little change in signal strengths or shapes compared to WT (Fig. 3) but slower decay, with the Y30F/Y39F variant showing the slowest decay at 8.5 s (Supplementary Fig. 6A). This suggests that Y30 and Y39 are involved in radical scavenging to some extent, but are not key players. Y121F, on the other hand, showed a clearly different behaviour compared to WT (Fig. 3, green spectra). While the formation and decay of radical features had similar velocities to WT, the tryptophanyl region was noticeably enhanced (12% maximum formation compared to 3% in WT), whereas the tyrosyl feature was virtually absent compared to WT. Y121 is therefore likely to be the major contributor to the strong tyrosyl feature observed in WT and a key station on a possible hole hopping path. It is, however, intriguing that mutation of this tyrosine did not affect the rates of formation and decay of the tryptophanyl feature.

Y121 is a buried residue and, thus, does not seem capable of efficiently transferring holes to external acceptors. In this regard, it has been proposed that hole hopping involving tyrosine side chains is promoted by the presence of a nearby appropriate proton acceptor, which could be the Nδ or Nε from the imidazole ring of a histidine if within 4Å[1]. In *Sm*AA10A, the hydroxyl group of Y121 is 2.7 Å away from the Nδ of solvent-exposed H160, which could feasibly act as a proton acceptor and provide a path to the solvent. Inspection of the crystal structure of *Sm*AA10A (2bem) showed that H160 is the only amino acid on the surface that can participate in a hole transfer route connected to Y121-O˙. Strikingly, the radical formation and decay characteristics of H160F are very similar to those of Y121F, including clear suppression of the tyrosyl feature (Fig. 3, purple spectra and Supplementary Fig. 7). Moreover, measurements with the Y121F/H160F variant showed similar results (Supplementary Fig. 6B). These observations not only strengthen the idea that Y121 is a major contributor to the WT tyrosyl feature but also show that formation of Y121-O˙ depends strongly on nearby H160.

The seemingly central role of the Y121/H160 couple in radical dissipation is supported by the strengthened tryptophanyl feature in variants where this couple is disrupted, with the fraction of W˙ detected increasing from 3% in WT to 12 % in Y121F, 11% in H160F and 20 % Y121F/H160F. In light of this central role, it is somewhat unexpected that the tryptophanyl feature shows similar velocities of formation and decay for WT, Y121F, H160F and Y121F/H160F. This suggests that precluding Y˙ formation in position 121 does not greatly influence the rate

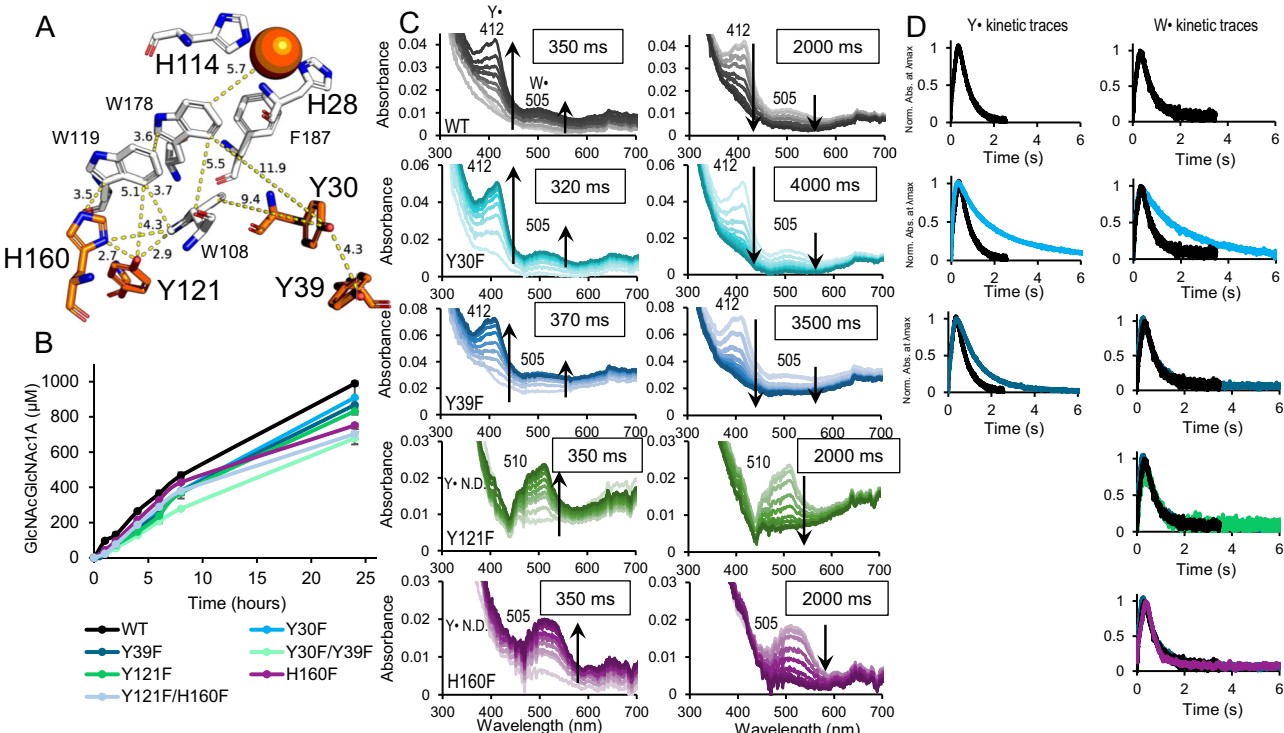

**Fig. 3 | Effects of mutations Y30F, Y39F, Y121F and H160F on LPMO performance and radical formation and decay.** WT (black), Y30F (light blue), Y39F (dark blue), Y121F (green) and H160F (purple) are shown. **A** Cu-site, aromatic redox core and two additional tyrosines, Y30 and Y39, in *Sm*AA10A (pdb 2bem). **B** oxidation of chitin over time in reactions containing 1 μM LPMO, 10 g·L⁻¹ chitin and 1 mM ascorbate performed at 37 °C in 50 mM Tris, pH 8.0, represented as mean values ± s.d. (*n* = 3; see Supplementary Fig. 4A for an overview of time-courses for all enzyme variants). **C** UV-Vis spectra obtained by stopped-flow absorption spectroscopy for reactions containing 75 μM LPMO-Cu(I) and 20 molar excess H₂O₂, at 4 °C and in 50 mM Tris, pH 8.0. Features were assigned as in Fig. 2. The left panels with spectra show formation until the maximum signal was detected and the right panels with spectra show the decay from maximum signal until resting state, with the times needed to reach maximum signals and full decay indicated in boxes. For the WT, these times are 350 ms and 2000 ms, respectively. Features which could not be detected are indicated by N. D. Spectra for variants with two mutations are shown in Supplementary Fig. 6. Further spectral details for H160F at longer time scales are shown in Supplementary Fig. 7. **D** kinetic traces for the tyrosyl or tryptophanyl radicals detected in each variant. WT traces are shown in each panel for reference. Figure 6 provides a summarizing overview of the formation and decay velocities of various enzyme variants.

of W˙ formation or decay. Two hypotheses could explain this behaviour: (i) Phe in position 121 is still able to scavenge radicals from the core Trp chains without accumulating a UV-Vis signal that can be detected with our experimental set-up; (ii) transfer from the core Trp chains to surface scavengers proceeds by alternative hole hopping pathways that avoid position 121. One possible scenario entails dissipation via Y30 and/or Y39. Such a scenario is supported by the reduced decay rate of the Y30F/Y39F variant that lacks both tyrosines (Supplementary Fig. 6A), but is contradicted by the lack of a Y˙ signal in the Y121F and H160F variants. To support this scenario, one would have to assume that formation and decay of Y˙ in positions 30 and 39 are too fast to be detected. The radical at Y121 may be more observable due to its stabilization by H160; such stabilization is not possible for Y30 and Y39, which lack a neighbouring histidine and are more solvent-exposed than Y121.

Activity assays using apparent "monooxygenase conditions" showed that all variants with mutations in Tyr and/or His positions perform similar to the WT, although there seems to be some reduction in activity (earlier plateauing) after 8 h for H160F and Y121F/H160F (Fig. 3, bottom left). So, interestingly, while mutations of these tyrosine and histidine residues have effects on radical formation and decay, their effects on enzyme stability are less pronounced compared to the effects of mutating W178 closer to the copper.

## Control experiments: mutational effects on temperature stability, substrate binding and oxidase activity
Given that most of the mutations in *Sm*AA10A have a negative effect on packing in the protein core, we measured the apparent melting temperature, Tm, to assess whether observed mutational effects could in part be due to destabilization of the protein fold. As expected, most variants show reduced stability relative to the WT enzyme (Supplementary Fig. 8). However, even the lowest Tm observed (56 °C for W178Y/W119F, compared to 71 °C for WT) is well above the maximum reaction temperature used in this study (37 °C), indicating that in every condition used the LPMO variants were well folded and stable. Of note, the W119F variant, which is discussed extensively below, was almost as stable (Tm = 70 °C) as the WT enzyme.

Substrate binding has been shown to affect LPMO stability, since poor binding increases the chance of potentially damaging futile turnover of available $H_2O_2$[50,51,68]. At the same time, stopped-flow amino acid radical detection was performed in the absence of substrate. To be able to reliably correlate radical formation to LPMO stability under turnover conditions, it is important to know the substrate-binding properties of the enzyme variants. Binding studies with all the variants in their Cu(I) state generally showed small differences (Supplementary Fig. 9). The W178Y/W119F variant shows reduced, but still considerable binding and has the largest difference compared to WT. Of note, all the variants were purified in one single step using chitin beads, indicating good binding to chitin.

In addition, we measured the oxidase activity, i.e. reductant-driven generation of $H_2O_2$ in a reaction without substrate[69,70], of all the variants, because variation in this activity could affect LPMO activity and stability in reductant-driven reactions (as in the lower left panels of Figs. 2 and 3). All variants showed low oxidase activity, which is common for AA10 LPMOs and contrasts the higher activities commonly observed for AA9 LPMOs[70]. Importantly, the background rate of $H_2O_2$ production through enzyme-independent oxidation of the reductant dominated overall $H_2O_2$ production for all variants, which showed oxidase activities similar to that of the WT (Supplementary Fig. 10).

## A closer look at the oxidative stability of hole hopping variants
To further explore the connection between radical formation and LPMO performance, we carried out activity assays under intentionally damaging conditions. In this case, reactions were carried out under "peroxygenase conditions", which implies using only priming amounts

of reductant (50 μM ascorbate), while $H_2O_2$ is added to the reaction at $t = 0$. Such reactions are much faster than the (reductant-driven) reactions done under apparent "monooxygenase conditions" that are depicted in Figs. 2 and 3, but will lead to increased enzyme inactivation if amounts of added $H_2O_2$ are high, as was the case here. Figure 4A shows the performance of WT and eight selected variants acting on chitin in reactions with 300 μM added $H_2O_2$, the latter being a damaging high amount for this enzyme acting on this substrate[42]. With two exceptions, Y121F and W119F, the variants all show lower product formation than WT, which, together with the shape of the progress curves, i.e., earlier plateauing, indicates increased enzyme inactivation. Despite the notable difference in radical formation discussed above, Y121F performed similarly to WT, while both H160F and Y121F/H160F, with similar changes in radical formation relative to WT, showed lower product formation and signs of increased enzyme inactivation. Strikingly, W119F differs from the WT and all the other variants in that it, under these conditions, displays a clear increase in initial activity, next to faster inactivation. As a consequence, W119F produces almost as much product as WT, but with a quite different progress curve (Fig. 4A). We elaborate on this remarkable finding below.

We also performed a peroxidase assay with 2,6-dimethoxyphenol (2,6-DMP), using increasing initial $H_2O_2$ concentrations. In this reaction, added $H_2O_2$ is consumed in a peroxidase reaction that results in the formation of coerulignone, a compound that absorbs visible light (2,6-DMP → hydrocoerulignone → coerulignone). The substrate in this reaction is far from optimal and not a natural LPMO substrate. This should imply that the enzymes are particularly vulnerable to damaging off-pathway reactions with $H_2O_2$, which again could mean that mutations in protective hole hopping pathways have noticeable effects. As expected, the rate of the peroxidase reaction, ΔAbs/min, increases with the $H_2O_2$ concentration, and this increase stops when $H_2O_2$ concentrations become damaging (Fig. 4B). At higher $H_2O_2$ concentrations enzyme activation is expected to occur increasingly early and this was reflected in a shortening of the linear portions of the product formation curves that were used to calculate the rates. Figure 4B shows that the WT enzyme could handle the highest $H_2O_2$ concentrations compared to the other enzyme variants. All hole hopping variants tolerate $H_2O_2$ less well and W178F and W178M, the only two variants in which hole transfer via position 178 likely is fully blocked, stand out by being extremely vulnerable: the peroxidase activity was almost completely lost even at low $H_2O_2$ concentrations. Variants with a Tyr in position 178 tolerate $H_2O_2$ better than W178F or W178M but still clearly less well than the WT. The other variants, including W119F, Y121F, H160F and Y121F/H160F tolerate $H_2O_2$ better that the W178Y containing variants but less well than WT (Fig. 4B).

Finally, we analysed β-chitin oxidation with controlled continuous dosing of $H_2O_2$ using glucose oxidase (GlcOx) for four selected enzyme variants (WT, W119F, W178Y and W178Y/W119F). GlcOx will engage in an oxidative reaction with glucose (which is added to the reaction), producing $H_2O_2$ as a byproduct[41]. This multi-enzymatic setup has been used before to fuel LPMOs in a controlled way[41,71]. At lower GlcOx concentration, i.e., low supply of $H_2O_2$, all enzyme variants displayed linear product formation curves (Fig. 4C), with W119F notably producing slightly more oxidized products than WT. At a 4-fold higher GlcOx concentration, i.e., higher supply of $H_2O_2$, the reactions were (expectedly) faster, whereas enzyme inactivation and differences between the four enzyme variants became more apparent (Fig. 4D). In this second setup, product formation started to plateau between 20 – 30 min of reaction, but with clear differences between the variants that are reflected in the shapes of the progress curves and product levels at the end of the reaction: WT clearly tolerates a higher dosage of $H_2O_2$ than W119F, whereas W178Y and W178Y/W119F are the most vulnerable. These results underpin the notion that a well-preserved hole-hopping route is necessary to tolerate damaging conditions, and that

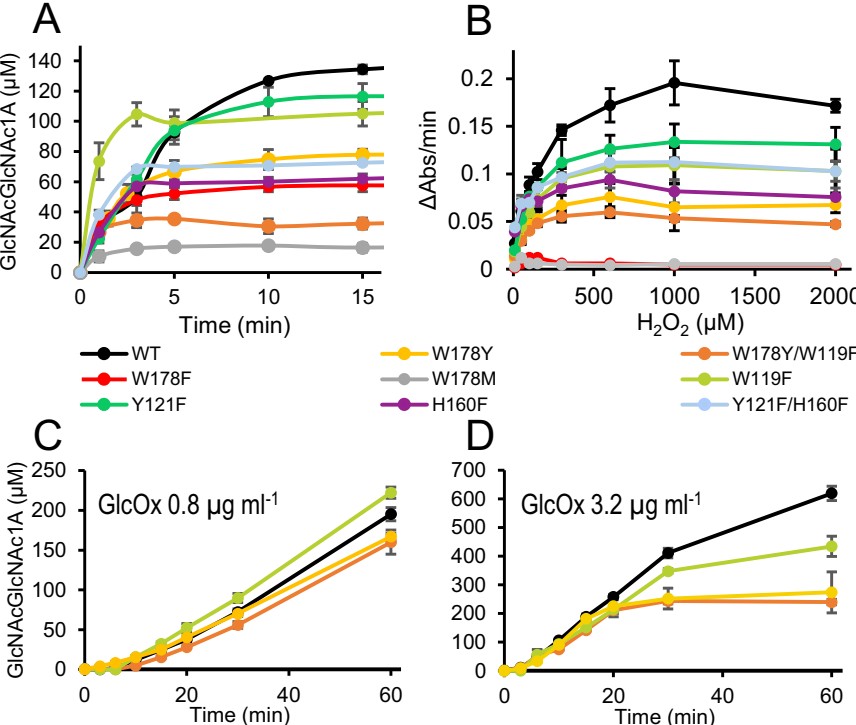

**Fig. 4 | Peroxygenase and peroxidase activity of WT *Sm*AA10A and variants.** WT (black), W178Y (yellow), W178Y/W119F (orange), W178F (red), W178M (grey), W119F (lime), Y121F (green), H160F (purple) and Y121F/H160F (light blue) are shown**. A** Oxidation of 10 g L⁻¹ β-chitin in reactions with priming (50 μM) amounts of ascorbate and 300 μM H₂O₂ in 50 mM Tris, pH 8.0. Data for all enzyme variants are provided in Supplementary Fig. 4B. **B** Dependency of the (linear) initial rate of an LPMO peroxidase reaction with 2,6-dimethoxyphenol (2,6-DMP)[92] in 50 mM Tris, pH 7.0, on the initial concentration of H₂O₂. **C, D** Oxidation of 10 g L⁻¹ β-chitin with 1 mM ascorbate in reactions including 15 g L⁻¹ glucose and 0.8 (C) or 3.2 (D) μg mL⁻¹ of GlcOx, in 50 mM Tris, pH 8.0. The LPMO concentration was 1 μM in all reactions with chitin, and 2 μM for 2,6-DMP oxidation. All data are represented as mean values ± s.d. (*n* = 3).

perturbing such a route at different points affects enzyme performance in different manners.

Taking our peroxygenase and peroxidase results, it seems safe to conclude that the observed mutational effects on radical formation and enzyme operational stability are largely, or even exclusively, due to variation in hole hopping. The data show that W178 is crucial for protection of *Sm*AA10A, and that all other residues tested contribute to such protection.

**Further analysis of the increased peroxygenase activity of W119F**

Intrigued by the increased initial activity on β-chitin of W119F, we explored the possibility that less efficient hole hopping in this variant, as demonstrated by less tyrosyl radical formation and three times slower radical decay (Fig. 2), leads to faster catalysis. MALDI-TOF MS of product formation showed identical spectra for WT and W119F (Supplementary Fig. 11), showing that the mutation, which is far away from the substrate-binding surface, does not affect the enzyme specificity.

Detailed kinetic characterization of the two enzyme variants using a recently developed rotating-disc electrode for monitoring H₂O₂ consumption[72] and various reaction conditions, showed that W119F indeed is a faster enzyme than WT, as evidenced by a higher rate of H₂O₂ consumption in the initial phase of the reaction with chitin (Fig. 5A; more data and discussion in Supplementary Fig. 12). While W119F consistently showed a higher initial rate than WT, the time traces for this variant (Fig. 5A, Supplementary Fig. 12) begin to flatten earlier, indicating faster self-inactivation. These trends align well with the results of the chitin degradation assays depicted in Fig. 4A, showing that the variant is faster but less well protected against the harsh conditions, i.e., high levels of H₂O₂, used in these assays. Of note, W119F also showed a small but noticeable increase in the rate of the off-

pathway peroxidase reaction (Supplementary Fig. 12). Taken together, the data show that the increased sensitivity of W119F for autocatalytic inactivation, which is demonstrated in multiple experiments above, is accompanied by a higher initial catalytic activity.

To ensure that the observed differences indeed are due to hole hopping effects, we evaluated possible impacts of the mutation on copper reactivity. Stopped-flow spectrofluorometric analysis of the kinetics of LPMO reduction by ascorbate and reoxidation by H₂O₂ (Fig. 4B, C, respectively) showed that WT and W119F have very similar reduction rates ($5.9 × 10^5 M^{-1} s^{-1}$ vs $4.5 × 10^5 M^{-1} s^{-1}$, respectively) and reoxidation rates ($7.9 × 10^3$ vs $8.8 × 10^3 M^{-1} s^{-1}$). For comparison, mutation of W178, which is much closer to the copper, did have effects on copper reactivity, showing 3 to 4-fold lower reoxidation rates (Supplementary Fig. 12E, F), pointing to secondary roles of this residue beyond hole transference.

Additionally, continuous-wave (CW) X-band Electron Paramagnetic Resonance (EPR) spectra of WT and W119F were collected to compare the geometric and electronic properties of the active sites (Fig. 5D and Supplementary Figs. 13 and 14). The two enzymes give similar spectra (Fig. 5D) showing a mixture of two subcomponents, one of which can be described with a rhombic **g**-tensor, while the other one exhibits an axial spectrum. There have been numerous examples of other AA10 LPMOs showing a similar behaviour[73–77], that is attributed to differences in the number of aquo/hydroxo species, $nH_xO$, coordinated ($n = 1,2$) to the copper center (see SI for more information). Importantly, the identical EPR parameters, particularly the resolved nitrogen hyperfine, evidences identical 3N1O coordination environments for both WT and W119F.

All in all, the data presented above, show that that the main difference between WT and W119F is a slowed down hole hopping path, which, as shown in Fig. 4 and 5 and in Supplementary Fig. 12, leads to

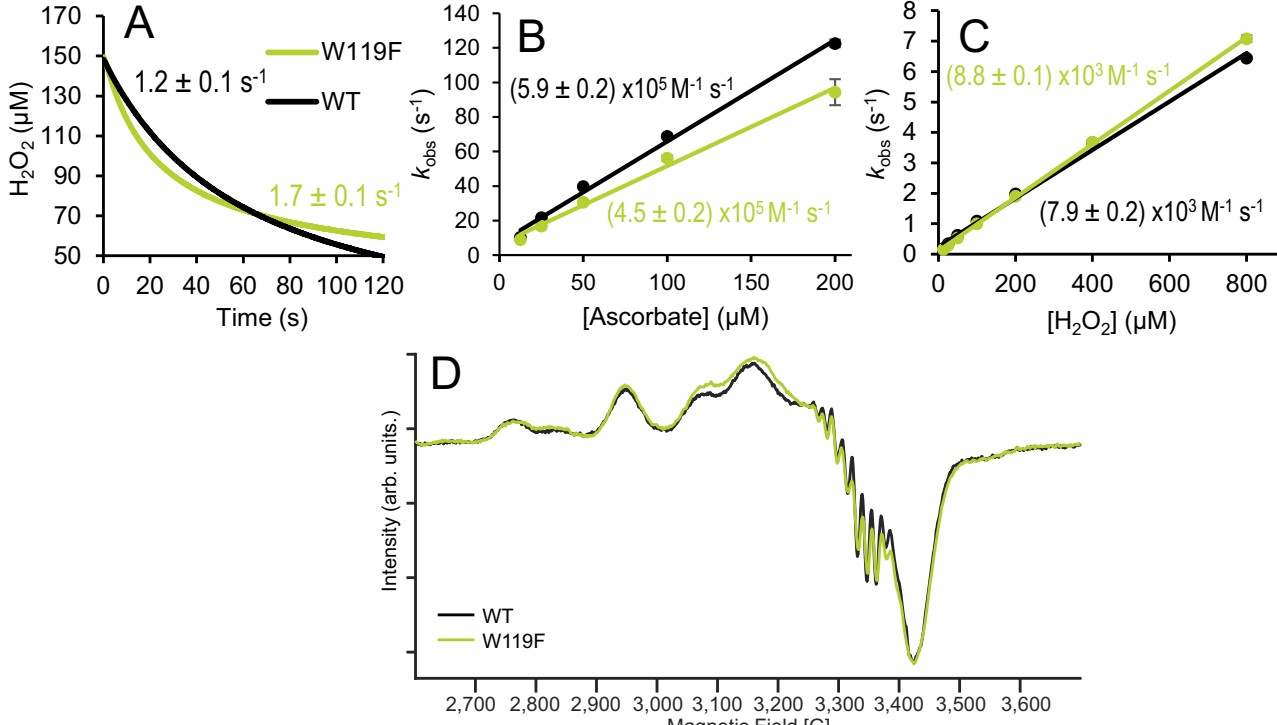

**Fig. 5 | Catalytic, redox and EPR properties of WT (black) and W119F (lime).**
**A** $H_2O_2$ consumption during the first 2 min of a reaction with chitin, measured using a rotating disc $H_2O_2$ sensor. The initial rates of $H_2O_2$ consumption, derived by linear regression of the first 15 s of reaction and corrected for substrate-dependent background and off-pathway peroxidase activity are indicated in the panel. The reaction mixtures contained 1 μM LPMO, 10 g L$^{-1}$ β-chitin, a priming amount (50 μM) of ascorbate and 150 μM $H_2O_2$ in 50 mM Tris, pH 8.0 and were incubated at 37 °C.
**B**, **C** Transient-state kinetics for reduction of LPMO-Cu(II) by ascorbate (**B**) and reoxidation by $H_2O_2$ (**C**). The derived second order rate constants for reduction and oxidation are indicated in the panels. The reactions were performed in 50 mM Tris pH 8.0 at 25 °C. Data are represented as mean values ± s.d ($n = 3$). **D** CW X-band (~9.46 GHz) EPR spectra of WT and W119F at 100 K. Intensity is expressed in arbitrary units (arb. units).

increased peroxygenase activity. EPR analysis of several of the other variants discussed above (Supplementary Figs. 13 and 14) also showed parameters identical to those of the WT, underpinning that the observed mutational effects do not relate to structural or electronic alterations in the copper site.

## Discussion

This work represents the most complete characterization of a hole hopping route in LPMOs to date and is the first to provide insight into the apparently elaborate hole hopping networks that may occur in bacterial AA10-type LPMOs (Fig. 6). The results provide insight into how holes may travel through proteins and how the nature and efficiency of the transfer routes affect enzyme functionality. Our sequence space analysis shows clear differences between phylogenetically distant AA10 LPMOs. While chitin-active LPMOs (such as SmAA10A) possess a conserved chain of Trp/Tyr from the copper site to the opposite surface, cellulose-active or mixed-activity LPMOs contain a larger fraction of redox-inactive amino acids (Phe, Val and Leu). Most importantly, we show here that these differences may not only affect redox stability but also catalytic efficiency. It is conceivable that having a less efficient hole hopping route may be beneficial for certain LPMOs acting on certain substrates in a certain ecological niche. The levels of available $H_2O_2$ in such niches likely plays a role.

Studies of cellulose-active ScAA10C, which lacks redox-active residues equivalent to W119, W108 or H160, have shown that oxidative damage accumulates in the catalytic histidines and in aromatic amino acids in the substrate-binding surface, close to the copper site[41]. This indicates that holes may be transferred to these residues. Cellulose-active AA10 LPMOs tend to have more hydrophobic and aromatic substrate binding surfaces compared to their chitin-active

counterparts[78]. It is conceivable that during evolution the incorporation of such amino acids improved cellulose-binding while also helping to scavenge damaging radical intermediates, considering also that tighter binding to the substrate prevents damage[50]. It is worth noting that our sequence analysis revealed possible connections between the presence of certain redox-active residues and the presence of a CBM. As an example, the cellulose-active and mixed-activity clades show a dichotomy for the position that is analogous to W119 in SmAA10A: single domain proteins have a Phe while multi-domain LPMOs contain a Trp.

Here, the critical role of strictly conserved W178 in AA10 LPMOs is demonstrated for the first time. One of the main differences in the copper coordination sphere of AA10s and AA9s is the amino acid in the proximal axial position. While AA9s contain a conserved Tyr, this residue is a Phe (e.g., F187 in SmAA10A; Fig. 1) in the majority of AA10s (except for ~10% of the proteins analysed here, that have a Tyr). For the Tyr in AA9s, radical formation has been observed[43,53,79,80]. Our data show that, in terms of possible hole hopping routes, W178, and not F187 is the analogue of the conserved Y in AA9s. Replacing W178 for a Phe or a Met (aromatic but redox inactive or redox-active but non-aromatic, respectively) abolished radical formation, implying that no other amino acid can act as hole acceptor and compensate for the absence of W178. Despite still being able to form radicals in the core, variants containing the W178Y mutation appear to have a heavily affected oxidative stability (Fig. 4A). On a side note, we detected an impact of W178 on the reaction of the reduced LPMO with $H_2O_2$ in the absence of substrate (Supplementary Fig. 12F), underpinning the importance of this residue for catalysis in AA10 LPMOs.

Mutagenesis of position W119 showed particularly intriguing and interesting effects. Blocking hole transference through the W119F

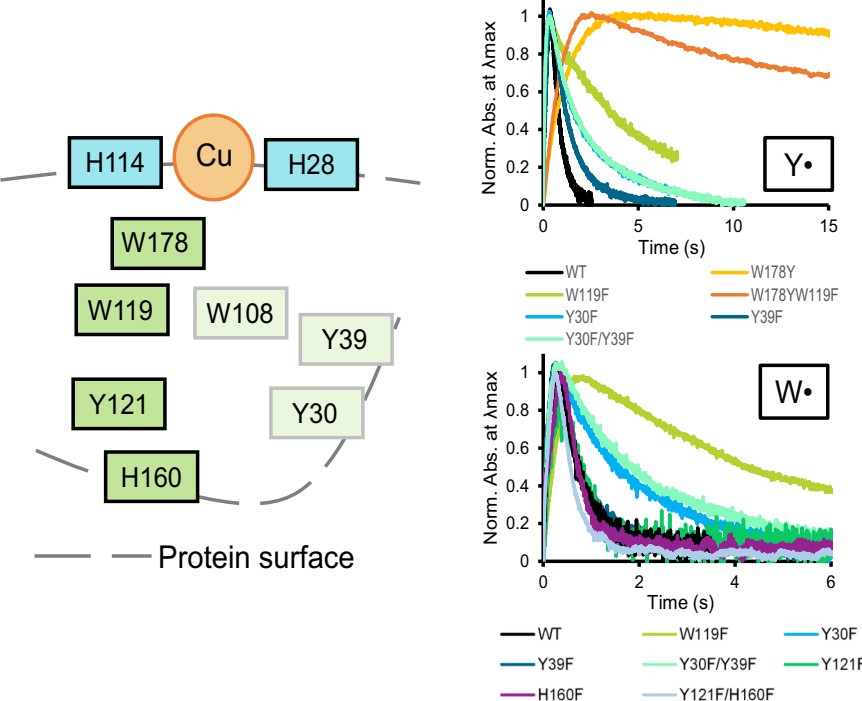

**Fig. 6 | Summary of radical formation and dissipation in *Sm*AA10A.** The left panel shows a schematic representation of the His-brace (H114-Cu-H28) and the amino acids positions targeted in this work, with those more relevant for function highlighted in darker green. Amino acids in the protein's surface are indicated by touching the grey line. The right panels show the accumulation and decay of Y˙ and W˙ radicals in *Sm*AA10 variants carrying mutations of the residues highlighted in the left panel. Some proteins show both radicals (WT (black), W119F (lime), Y30F (light blue), Y39F (dark blue), Y30F/Y39F (light green)), only Y˙ (W178Y (yellow), W178Y/W119F (orange)) or only W˙ (Y121F (dark green), H160F (purple), Y121F/H160F (light blue)). Note that W178F and W178M are not shown since they are UV-Vis silent.

mutation resulted in accumulation of W˙ in W178 and a slower decay of these radicals. At the same time, the formation of downstream Y˙ (mostly Y121, as shown here) was hampered. Interestingly, all measurements of activity, by quantification of soluble oxidized products or state-of-the-art detection of $H_2O_2$ consumption in real-time showed that the W119F variant has higher initial peroxygenase activity. Thus, by inhibiting a competing hole hopping route, use of the hole for productive catalysis is promoted, albeit at the cost of enzyme redox stability. So far, this competition between inactivation, activity and protection has been only hypothesized[12,53], and our engineered LPMO would be the first example showing experimentally that redox biocatalysts can be made more active by slowing-down hole transfer. Notably, not all AA10 LPMOs contain a Trp at position 119 (Fig. 1). This natural variation may reflect adaptation to a variety of environments in terms of substrate and $H_2O_2$ concentrations, putting different evolutionary pressures on the trade-off between redox stability and catalytic efficiency.

The effects of "downstream" mutations (Y30, Y39, Y121, H160), alone or in combination, indicate redundancy in the hole-hopping routes of *Sm*AA10A, and such redundancy has been observed in other oxidoreductases[1] (Fig. 6). The results clearly show that blocking hole hopping at individual "downstream" points in the route does not shut down transfer. For example, mutation of surface-exposed Y30 and Y39, led to slower radical decay and somewhat reduced enzyme stability but did not have as dramatic effects as mutation of W178. Mutations of the Y121-H160 pair showed that the prominent tyrosyl signal corresponds to Y121 and that formation of an observable radical at this position depends on H160. Still, mutations of Y121 and H160 did not slow down decay of the tryptophanyl signal, which, apparently can dissipate via routes that do not involve the Y121-H160 pair and other, detectable, radical intermediates. This could be routes involving Y30 and Y39, which likely are easily deprotonated since their hydroxyl

groups are solvent exposed and show associated water molecules in the crystal structure of the enzyme.

Supplementary Figure 15 provides a closer look of the environment of W119 and the Y121-H160 pair. It is worth noting that the indole group of W119 has several crystal waters nearby and is at hydrogen bonding distance of the carboxylic head group of an aspartate (D149). These nearby groups will facilitate proton transfer associated with oxidation of W119. Although it is not possible to pinpoint exactly how holes are removed from the protein after having traversed its core, reaching W119, it is clear that surface-located residues must be essential. The Y121-H160 pair, in which the Nδ of H160 hydrogen bonds (2.7 Å distance) with the hydroxyl of Y121, is of particular interest (see Supplementary Fig. 15). Mutation of these residues in *Sm*AA10A led to decreased enzyme redox stability, showing that stability suffers if an escape point for radical transfer is removed. Radical transferring Tyr-O˙---HN⁺-His pairs have been characterized in other redox systems, including enzymes and photosystem II (PSII) of green plants and cyanobacteria[59,81]. Considering that Y121 seems responsible for the tyrosyl signal, the effect of the single Y121F mutation on enzyme stability under redox stress is modest, whereas the effect of the H160F mutation is more pronounced. This suggests that hole transfer from core Trp residues to the solvent exposed H160 can occur, bypassing Y121. As shown in Supplementary Fig. 15, H160 is part of a cluster of redox active amino acids (W119 and W108; and, in the wild-type, Y121) that also contains crystal waters close to W119 and H160. Thus, there may be multiple options for hole and associated proton transfer and for the generation of partially delocalized structures that facilitate transfer.

One possible caveat in our approach is that we link enzyme stability in reactions where the protective effect of the substrate plays a role with the detection of Y• and W• radicals in reactions that lack substrate, where inactivation is more likely to occur[41]. The

question then arises whether the inactivation observed during chitin turnover is due to the unbound fraction of LPMO in solution being damaged or whether inactivation may also result from holes going astray in what in principle is a productive enzyme-substrate complex. Phrased differently, did hole hopping routes in LPMOs evolve to protect the unbound fraction of LPMO? Better protection of unbound LPMOs might explain why some *Sm*AA10A variants with altered hole hopping routes (notably W119F and Y121F) perform like WT (or better, for W119F) under redox stress with chitin (Fig. 4A) but are clearly inferior to WT in reactions with the small molecule 2,6-DMP (Fig. 4B), where substrate protection is not expected. Most importantly, our results with W119F show that hole hopping also plays a role in turnover conditions: the fact that inhibition of hole hopping leads to higher initial activity must mean that, in the wild-type enzyme, a certain fraction of holes dissipates from the catalytic center while the substrate is bound.

In conclusion, using a combination of experimental techniques we have identified putative complex and long-range hole hopping routes in AA10-type LPMOs and we have dissected one such route in *Sm*AA10A. The results show a correlation between the efficiency of radical formation and protection against oxidative damage. Aromatic residues close to the copper site are most important, while "downstream" dissipation routes are more diffuse, perhaps showing redundancy. We show also the importance of a Tyr-His pair in the surface of LPMOs for radical formation and dissipation. It is interesting to note that LPMO catalytic domains generally are very small enzymes, typically 150 – 200 residues (170 for *Sm*AA10A). Our data show that a considerable fraction of these small proteins has evolved to deal with the highly damaging redox species that are generated during productive and non-productive catalysis. Finally, and most importantly, our data provide experimental evidence for the idea that slowing down hole hopping can boost enzyme activity. This sheds light on the activity/stability trade-off, which, considering the large sequence diversity, must vary considerably among naturally occurring LPMOs.

## Methods

### Sequence space and phylogenetic analysis of LPMOs cores

AA10 LPMO sequences were retrieved from the dbCAN2 database[82] (08062022 version), which performs an automated annotation of carbohydrate-active enzymes in available genomes. Three sequence datasets were generated using the in-house script dbcan_curation.sh: one for all AA10 LPMOs in dbCAN2, a second one for LPMOs with only a catalytic domain (CD) and a third one for LPMOs with an extra domain known as a carbohydrate binding module (CBM). The signal peptides were removed in all datasets, leaving the copper-binding N-terminal histidine as the first amino acid. Three phylogenies (one per dataset) were built using fasttree[83] after sequence alignment with MAFFT (L-INS-i option)[84] (Supplementary Fig. 1). AA10 LPMOs can be functionally annotated and clustered according to the three known types of substrate specificity they have: chitin-active, cellulose-active or mixed chitin-/cellulose-active. In the phylogenetic studies, these activities appear in two well differentiated clades, one for chitin-active LPMOs and one for LPMOs with cellulose and mixed chitin/cellulose activity (clades A and B in Supplementary Fig. 1, respectively). The corresponding sequences of clades A (for chitin-active) or B (for cellulose/mixed activity) were manually selected from each of the three phylogenies built, and (combinations of) key residues in the protein core were analysed for each of these after re-alignment of sequences with MAFFT (L-INS-i option). By doing so, chitin active^CD vs chitin active^CBM vs cellulose/mixed activity^CD vs cellulose activity^CBM datasets were compared. Based on conservation patterns, residues possibly participating in a hole hopping route were considered using LPMO10A from *Serratia marcescens* (*Sm*AA10A, also known as CBP21) as a starting point. This chitin-active LPMO is one of the best characterized LPMOs and

its core is particularly rich in aromatic residues. The positions analysed were Y30, Y39, W108, W119, Y121, H160, W178 and F187 (numbering starting from H28 as first amino acid after signal peptide cleavage). Note that six of these eight positions are tyrosine or tryptophan in *Sm*AA10A. Upon further consideration, Y30 and Y39 were discarded from the analysis of pattern conservation given their relatively high variability in AA10s and based on preliminary experimental results, showing their limited relevance compared to the other six residues. The script to analyse combinations of amino acids in the AA10 LPMOs sequence space (count_amino_acid_combinations.py) can be used for any protein family. The scripts used in this work are publicly available at https://github.com/IAyuso, where they are described.

### Design of enzyme variants with Rosetta

The crystal structure of *Sm*AA10A (2bem, chain C) was used to obtain the best candidate enzyme variants carrying single mutations of Y30, Y39, W108, W119, Y121, H160, W178 or carrying double mutations of Y30/Y39, W119/W178 and Y121/H160. An initial relaxation step was performed on the pdb structure (2bem) to optimize side chain configurations and minimize clashes, with a constraint to starting coordinates to minimize large deviations[85], and including 5 relaxation cycles and 10 structures generated. The structure with minimum total energy among the relaxed structures obtained was selected for further design. The fixbb protocol was then run on the relaxed structure, varying either single or double positions, to find the least destabilizing mutations[86]. Extra samples of the Dunbrack rotamer library were specified for buried aromatic residues (EX option). 100 independent runs were collected and ranked by total energy, and the best (usually unique) solution was selected. In the case of W108, a second run generating another 100 models was performed excluding A as an option, since it was the unique solution of the first run and we wanted to screen more options in this position. To consider the copper in the simulations, the sodium atom in 2bem was replaced with a copper atom[57], and the flag –auto_setup_metals was included in all steps.

### Mutagenesis, cloning, expression, and purification of *Sm*AA10A variants

The wild type *Sm*AA10A and all its variants were expressed and purified as described previously[17]. 15 variants were designed targeting the positions described above: Y30F, Y39F, W108Y, W108M, W108F, W108A, W119F, Y121F, H160F, W178Y, W178M, W178F, and the double mutants Y30F/Y39F, W119F/W178Y and Y121F/H160F. Plasmids expressing the variants were generated by site directed mutagenesis using QuickChange (Agilent), with the primers shown in Supplementary Table 1, except for the W108M, W108Y, W178M and W119F/W178Y variants, for which gene fragments were synthesized by Twist Biosciences and cloned using the Gibson Assembly® kit (New England Biolabs) into the pRSET-B backbone (i.e., the same backbone as for all other enzyme variants). Successful cloning of 12 of the 15 variants was confirmed by sequencing. For the remaining 3, W108F, W108A and W108M, despite several cloning attempts and sequencing of many colonies per variant, undesired mutations were always detected, meaning that the correctly mutated protein variants could not be produced, possibly because of the leaky nature of pRSET-B expression construct and accumulation of misfolded protein. The 12 successfully constructed pRSET-B variants were transformed into One Shot™ BL21 Star™(DE3) or OverExpress™ C43(DE3) (Invitrogen) chemically competent *Escherichia coli*, which were used as expression hosts. The expression protocol was very similar for all enzyme variants: a single colony was used to inoculate 2 L flasks containing 1 L of LB medium supplied with 100 μg mL⁻¹ of ampicillin. The expression of LPMOs in pRSET-B is constitutive, and after inoculation the culture was grown for 16 h at 37 °C, 200 rpm, except

for W178Y, W178F and H160F, where expression was performed at 30 °C at 200 rpm. The cells were harvested, and each protein was collected by periplasmic extraction with cold osmotic shock. In the case of W108Y, several conditions (LB or TB, 30 or 37 °C) were tried, but soluble expression was not observed so this variant was discarded.

All the proteins were purified in one single step after loading the periplasmic extracts, supplemented with $(NH_4)_2SO_4$ to a final concentration of 1 M, onto a chitin resin column (New England Biolabs) equilibrated with 50 mM Tris pH 8.0, 1 M $(NH_4)_2SO_4$. Unbound protein was removed by extensive washing with the same buffer, and pure protein was eluted with 20 mM acetic acid. Eluates were immediately dialyzed vs 50 mM Tris, pH 8.0, and stored at 4 °C. Copper saturation was achieved by incubation with a 3-fold molar excess of $CuSO_4$ for 30 mins at room temperature, after which excess copper was removed by four successive rounds of dialysis vs large volumes of 50 mM Tris, pH 8.0, aiming for a $10^6$ dilution factor. The protein concentration was quantified by measuring the absorbance at 280 nm, using theoretical extinction coefficients.

Copper saturation of $Sm$AA10A variants for EPR experiments was either achieved through the protocol stated above, or through incubation with 3-fold molar excess of $CuSO_4$ for 30 mins on ice. Excess copper was subsequently removed by extensive ultrafiltration using Amicon centrifugal filters with a molecular mass cutoff of 10 kDa, with simultaneous exchange of buffer to 50 mM Tris buffer, pH 8.1.

**Stopped-flow UV-Vis detection of amino acid radicals**
All the experiments were carried out with a stopped-flow rapid spectrophotometer (SFM4000, BioLogic Science Instruments) coupled to a TIDAS® S 500 MCS UV/NIR 1910 (J&M Analytik AG) diode array for detection of UV-Vis traces. Double mixing experiments were used to generate UV-Vis traces after mixing LPMO-Cu(I) with $H_2O_2$. To do this, LPMO-Cu(II) (300 μM solution) was mixed with 1 molar equivalent of ascorbate (unless otherwise stated), followed by ageing the reaction for 10 s in a delay line to ensure formation of LPMO-Cu(I). After that, the Cu(I) protein was mixed with 20 molar equivalents of $H_2O_2$ and the UV-Vis traces were collected with a minimum of 1.5 ms sampling between spectra and with an instrument dead time of ~2.9 ms. All experiments were carried out at 4 °C in 50 mM Tris, pH 8.0, unless stated otherwise. All reagents were deoxygenated by $N_2$ sparging using a Schlenk line and subsequently prepared in sealed syringes in an anaerobic chamber. The stopped-flow rapid spectrophotometer was flushed with a large excess of deoxygenated buffer before coupling the sealed syringes and performing the experiments. The $H_2O_2$ concentration was determined by measuring absorbance at 240 nm and using an extinction coefficient of 43.6 $M^{-1}$ $cm^{-1}$. All experiments were done at least in duplicates.

**Stopped-flow fluorescence spectroscopy**
We used the differences in intrinsic fluorescence between the Cu(II) and Cu(I) states of $Sm$AA10A and its variants[87] to measure the kinetics of LPMO reduction by ascorbate and reoxidation by $H_2O_2$, as previously described[47]. All the experiments were carried out with a stopped-flow rapid spectrophotometer (SFM4000, BioLogic Science Instruments) coupled to a photomultiplier tube with an applied voltage of 600 V for detection. The excitation wavelength was set to 280 nm, and the fluorescence increase (for reduction) or decay (for reoxidation) was collected with a 340 nm bandpass filter. All experiments were carried out at 25 °C in 50 mM Tris buffer, pH 8.0. To monitor reduction of Cu(II) to Cu(I), LPMO-Cu(II) (5 μM final concentration) was mixed with different concentrations of ascorbate (ranging from 12.5 – 200 μM final concentrations). For reoxidation, double-mixing experiments were performed in two steps. In a first step, the LPMO-Cu(II) (10 μM initial concentration) was mixed with one molar equivalent of L-Cys for 10 s to form LPMO-Cu(I). In a

second step, the in situ generated LPMO-Cu(I) was mixed with different concentrations of $H_2O_2$ (ranging from 12.5 – 800 μM after mixing) to monitor the decay of fluorescence. All reagents and the stopped-flow rapid spectrophotometer were deoxygenated as described above. BioKine32 software (V4.74.2, BioLogic Science Instruments) was used to fit the data to the single hyperbola $y = at + b + c \cdot e^{-k_{obs} \cdot t}$. In all cases, plots of $k_{obs}$ vs [ascorbate] or [$H_2O_2$] were fitted using linear least squares regression to obtain the apparent second order rate constant of the reduction or oxidation step ($k_{1app}$ or $k_{2app}$, respectively) with SigmaPlot v14.0. All experiments were done at least in triplicates.

**Reactions with β-chitin and quantification of oxidized products**
The ability of $Sm$AA10A variants to oxidize β-chitin was evaluated with different reaction setups. All the reactions were prepared in 2 mL tubes and incubated at 37 °C, 800 rpm in an Eppendorf Comfort Thermomixer with a temperature-controlled lid. Initially, time-course reactions with no external source of $H_2O_2$ were prepared by incubating 10 g $L^{-1}$ β-chitin from squid pen (Batch 20140101, France Chitin), milled with zirconium oxide grinding tools in a PM200 planetary ball mill, (Retsch) to a particle size of 75–200 μm, and 1 μM LPMO in 50 mM Tris, pH 8.0, for 30 mins and then initiating the reaction by adding 1 mM ascorbate. Samples were taken at different time points over 24 h and filtered with a MultiScreen™ 96-well filter plate (Merck). Under these conditions, the LPMO reaction is limited by the rate of in situ generated $H_2O_2$[88].

A second setup involved the same reaction conditions but with different amounts of added $H_2O_2$ (0-1000 μM; added at $t = 0$) and with sampling at 0, 3 and 60 mins. The reactions were started by adding 1 mM ascorbate. A more detailed time course reaction was obtained using an intermediate $H_2O_2$ concentration, 300 μM, and only priming amounts of ascorbate (50 μM final concentration), with sampling at 0, 1, 5, 10, 15, 30 and 60 mins. All reactions were done in triplicates, and control reactions without ascorbate or without enzyme were included.

Soluble products in the filtered samples were degraded by overnight incubation at 37 °C with 1 μM chitobiase from $Serratia$ $marcescens$ ($Sm$CHB)[89]. $Sm$CHB will degrade the soluble products to GlcNAc (native monomer) and GlcNAcGlcNAc1A (chitobionic acid, the oxidized dimer). Analysis of chitobionic acid was performed using a 100 × 7.8 mm Rezex RFQ-Fast Acid H+ (8%) (Phenomenex) column operated at 85 °C in an RSLC system (Dionex), with isocratic elution with 5 mM sulphuric acid. 8 μL samples were injected and analytes were eluted during 6 min, using a 1 mL $min^{-1}$ flow rate. The analytes were monitored using a 194 nm UV detector. Quantification was done by creating a standard curve (25–800 μM) of chitobionic acid. The areas of the peaks corresponding to chitobionic acid were quantified using Chromeleon (v 7.0). The standards were generated by mixing 3 mM $N$-acetyl-chitobiose (Megazyme, 95% purity) with 0.1 mg $mL^{-1}$ of a chito-oligosaccharide oxidase from $Fusarium$ $graminearium$ in Tris pH 8.0 at 20 °C and overnight incubation for complete oxidation[89,90].

**Reactions with β-chitin using glucose oxidase as source of $H_2O_2$**
Release of soluble oxidised products from β-chitin (10 g $L^{-1}$) by $Sm$AA10A variants (1 μM) was also assessed in the presence of glucose (15 mM) and glucose oxidase from $Aspergillus$ $niger$ (GlcOx; type VII; Sigma-Aldrich). GlcOx is a FAD-dependent oxidase that converts glucose to $D$-gluconic acid with high specificity, reducing $O_2$ to $H_2O_2$ upon regeneration of the FAD cofactor[91]. The GlcOx concentrations used were 0.8 and 3.2 μg $ml^{-1}$ corresponding to ~4.2 and 17.6 μM $H_2O_2$ produced per minute, respectively, under the conditions of the assay. These values were obtained prior to the experiments using the Amplex Red/HRP method[69] using similar reaction set ups but excluding LPMO. 15 mM glucose was used as it was found to be approximately the minimum saturating substrate

concentration for linear production of $H_2O_2$ over 1 h with 3.2 µg ml$^{-1}$ GlcOx. Reactions were performed in 2 ml Eppendorf tubes in 50 mM Tris, pH 8.0, at 37 °C in a thermomixer at 800 rpm. Each reaction was started by addition of 1 mM ascorbate and stopped by passage of the reaction mixture through a 0.45 µm vacuum filter, followed by digestion of soluble products with 0.4 µM chitobiase from *S. marcescens* at 37 °C overnight.

## Peroxidase reactions with 2,6-dimethoxyphenol (2,6-DMP)

LPMO-catalysed oxidation of 2,6-dimethoxyphenol (2,6-DMP) and subsequent formation of coerulignone using $H_2O_2$ as co-substrate was measured as previously described[92] with some changes. Reactions were performed at pH 7.0 to avoid autooxidation of 2,6-DMP, which is considerable at pH 8.0. The reactions were run in 96-well plates in 100 µL volumes containing 50 mM Tris, pH 7.0, 2 mM 2,6-DMP and varying concentrations of $H_2O_2$ (0–2000 µM). Reaction mixtures without the enzyme were prewarmed at 30 °C for 5 min, and the oxidation of 2,6-DMP was started by adding the LPMO to a final concentration of 2 µM. The formation of coerulignone was monitored at 473 nm for 5 min using a Multiskan$^{TM}$ FC microplate photometer (Thermo Fisher) and normalized by subtraction of the absorbance signal from a negative control reaction without enzyme. The data was acquired using SkanIt RE (v7.0.2) and analysed in Microsoft Excel 365 MSO (v2311). All reactions were performed in triplicates. Initial velocities were calculated using the linear part of the progress curve and linear regression. The increase of $\Delta Abs^{473nm}$ in the initial, linear part of the reaction was used as velocity and plotted vs $H_2O_2$ concentration.

## Real-time monitoring of $H_2O_2$ turnover

The consumption of $H_2O_2$ by *Sm*AA10A and the W119F variant acting on 10 g L$^{-1}$ β-chitin was continuously measured using a $H_2O_2$ sensor technology recently described by Schwaiger et al.[72]. In short, the $H_2O_2$ sensor is based on a Prussian blue modified gold rotating disc electrode, which is rotated at an angular velocity of 50 s$^{-1}$ to achieve fast response times of <2 s, in a thermostated electrochemical reaction chamber (37 °C). The preparation of the $H_2O_2$ sensor via electrochemical deposition and its operation is described in detail in the Supplementary Methods.

## EPR spectroscopy

For EPR measurements of *Sm*AA10A and its variants, 0.5 mM to 2 mM solutions in 50 mM Tris at pH 8.1 were prepared and filled in 2.8 mm (OD) custom quartz EPR tubes. Continuous-wave (CW) X-band (~9.46 GHz) EPR spectra were recorded with a Bruker MS 5000 spectrometer at 100 K. Spectra were collected with the following parameters: field modulation frequency: 100 kHz; field modulation amplitude: 4 G; scan time: 120 s; effective time constant: 0.05 s; effective number of points: 1000; number of scans: 3 to 13. The EPR data was collected with ESRStudio (v1.74.3) and processed and analysed in Matlab R2023a and simulated using the EasySpin package (v. 6.0.0-dev.51)[93].

## Reporting summary

Further information on research design is available in the Nature Portfolio Reporting Summary linked to this article.

## Data availability

All data generated in this study have been deposited in the figshare repository under accession number https://doi.org/10.6084/m9.figshare.25261732. The crystal structure of *Sm*AA10A (2bem) can be found at https://doi.org/10.2210/pdb2BEM/pdb. The dbCAN database can be accessed and downloaded at https://bcb.unl.edu/dbCAN2/index.php.

## Code availability

All the code used for this study is publicly available at https://github.com/IAyuso.

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

## Acknowledgements

The authors gratefully acknowledge funding from the European Research Council (ERC) through a Synergy Grant (grant number 856446). LS and RL gratefully acknowledge funding from the Austrian Science Fund (FWF) for the Doctoral Program BioToP - Biomolecular Technology of Proteins (W1224-B09). GEC thanks the Max Planck Society for funding.

## Author contributions

I.A.-F. and V.G.H.E designed the study. I.A.-F., T.Z.E-M., J.H., O.G., K.R.H., L.S. T.S.M. and A.A.S. performed the experiments. L.S. & R.L. developed methods. I.A-F., T.Z.E.-M., J.H., O.G., K.R.H., L.S., A.A.S., R.L., G.E.C., M.S., Å.K.R and V.G.H.E. interpreted the data. I.A.-F., T.Z.E-M. and V.G.H.E wrote the initial manuscript. All authors contributed to revising and writing the final version of the manuscript.

## Competing interests
The authors declare no competing interests.

## Additional information

Iván Ayuso-Fernández or Vincent G. H. Eijsink.

**Peer review information** *Nature Communications* thanks Tony Vlcek and
the other, anonymous, reviewer(s) for their contribution to the peer
review of this work. A peer review file is available.

