## [Peer Review File · Nature Communications]

Mutational dissection of a hole hopping route in a lytic polysaccharide monooxygenase (LPMO)REVIEWER COMMENTS

Reviewer #1 (Remarks to the Author):

This is a very important manuscript that demonstrates the presence and functioning of protective hole-hopping pathways in an LPMO enzyme. Presumably, conclusions from this work could be extended and generalized to other oxidases. The authors have identified relevant tryptophan and tyrosine residues in the enzyme structure and established their functions by judicious mutations, replacing Trp and/or Tyr by inactive phenylalanine. It was shown that, in the absence of a substrate, the hole is channeled across an enzyme molecule, starting at a Trp residue close to the Cu center and ending at a surface Tyr/His pair by forming a deprotonated Tyr[•]. The manuscript makes a plenty of important points about enzyme activity and protection. The very existence of the hopping pathway was demonstrated, mutation of the first Trp residue (5.2 Å from the Cu center) in the pathway affects both protection and enzymatic activity while residues further down the pathway affect only the protection. The middle part of the hopping pathway exhibits some redundancy, whereby the holes can follow different paths toward the same Tyr/His target. It is shown the diminishing the protection activity enhances catalysis but at the expense of enzyme stability – pointing to a competition between substrate oxidation and enzyme protection by hole hopping. The manuscript is very well and clearly written, all results and conclusions are well documented. It is recommended for publication with only a few comments/suggestions:

It is recommended to state clearly the nature of the first hopping steps. Is W178 supposed to be oxidized by HO[•]?

W119, presumably the second hopping intermediate, seems to be even less solvent exposed than W178 (Fig. 1 – left). Taking guidance from photolyase, this would imply that W119 oxidation is energetically uphill (which is acceptable for an intermediate hopping step). How would W119^{•+} be deprotonated? By H160? Is there another proton – accepting residue in the vicinity of the W119 indole NH? Or, at least, a close-lying water molecule? Please comments on these issues.

The discussion of the kinetics of Y121F mutants (p. 13, also p. 24) remains open-ended. What is author's conclusion/preferred mechanism? Could oxidized Y30 and Y39 be readily deprotonated owing to their high water-exposure?

On p. 24, an interesting point is made about the W119-H160 pair. Histidine oxidation seems thermodynamically impossible. However, a partially delocalized structure involving also the W119 proton could be feasible. In the WT, H160 (which seems to be solvent-exposed) could perhaps play an intermediate role in the electron and proton transfer involving both W119 and Y121. Please comment. A detailed structural picture of this region would help.

p. 13, lines 350-355: This sentence is too long a hard to understand. It is recommended to rephrase it and (for comparison) state also the W[•] fraction detected for the WT (3% from p. 7?).

Reviewer #2 (Remarks to the Author):

“Mutational dissection of a hole hopping route that can be tuned to boost peroxygenase activity in LPMOs” by Ayuso-Fernandez et al describes work analysing hole/electron transfer pathways through SmAA10 (previously known as CBP21) using site-directed mutagenesis. Using a sequence analysis of the AA10 family and a structure guided approach the authors designed mutations in the core Trp and Tyr residues found in this family of enzymes near the enzyme active site to investigate whether they acted as protective hole hopping routes in the protein. Using stopped flow spectrophotometry they were able to observe the appearance of Trp and Tyr radicals in the protein in the absence of substrate and observed differences in the rate at which these radicals formed or decayed as a result of their mutations. Activity assays were also used to investigate the effects of the mutations on substrate activity and on peroxide sensitivity revealing some differences in the behaviour of the mutants compared to the wild type enzyme. Ultimately, the WT enzyme was the most effective enzyme but some useful insights into the potential importance of the core Trp/Tyr residues was revealed in allowing reactive species to be channelled away to the protein surface during unproductive reactions.

Overall, the paper is very well written and clear, the experiments have clearly been performed to a high standard and the data are well presented. For me, it would be nice to include the WT data for the stopped flow experiments in figure 2 as flicking back several pages to figure 1 for comparison was slightly unsatisfactory, particularly as we tend to read on screens these days.

Other than that minor suggestion, there were two main things that stood out to me that should be considered by the authors:

1. To me, the “boosting” of the peroxygenase activity in the W119F mutant is not that exciting. The increase in rate doesn’t seem huge and the follow up experiments show marginal differences compared to wild type. It does not result in an overall increase in product yield compared to WT and as the authors admit the enzyme becomes more rapidly inactivated so ultimately the WT enzyme is still the best one here. Whilst the observations with this mutant are interesting, I don’t think that there is that significant a “boost” in the

peroxygenase activity of the enzyme to suggest that it's going to be a really useful route towards engineering better enzymes for the future. I would therefore suggest putting a little less emphasis on this observation and consider changing the title slightly to something along the lines of "Mutational dissection of a hole hopping route through a bacterial LPMO." The results are still very interesting and there is no real need to "oversell" this.

2. My other concern is a technical one that relates to the consequences of introducing mutations into the hydrophobic core of a protein. The authors do not appear to have done anything to investigate the potential effects this could have on the fold of the protein more generally. I appreciate that control experiments using EPR suggest that there is little effect on the active site of the enzyme and the peroxygenase stability measurements show that many of the mutants are more sensitive to oxidative damage as one might expect. My question is, could this also result from a general destabilisation of the protein which could also limit the utility of the hole hopping pathways? The mutations may not necessarily simply block the hole hopping route and so some verification that the overall fold of the protein is unaffected would be a nice addition. A detailed structural analysis probing the protein dynamics in these mutants will undoubtedly be in the future plans of the authors and I am not suggesting that it's necessary to do that here. Some simple thermostability measurements could be made with the mutants to show that the mutations are not generally destabilising and that would just help to close off that question in terms of introducing mutations into the core of the protein.

Addressing this second point will just firm up the conclusions a little bit in what is a very good paper. I very much enjoyed reading the paper and I hope that the authors find these comments useful moving forward.

Reviewer #3 (Remarks to the Author):

This manuscript by Ayuso-Fernandez and co-workers characterizes variants of a copper-dependent lytic polysaccharide monooxygenase/peroxygenase from the AA10 family to understand the possible functional significance of tyrosine and tryptophan residues in controlling the potent chemistry leading to carbohydrate oxidative degradation. The thesis is essentially the idea espoused by Gray and Winkler that enzymes that make such potently oxidizing intermediates have built in fail-safe mechanisms – specifically networks of aromatic residues to pump oxidizing equivalents or "holes" to the protein surface where they can be quenched by any reductant – to make them more robust catalysts. The work uses analytical chemistry, kinetic data, and some spectroscopy to identify a primary hopping pathway that, somewhat surprisingly, seems to trade a modest diminution in efficiency for a substantial gain in robustness. I think the work has value and is generally well done but can be improved with some revisions.

My recommendations:

1. The most fascinating case of functional hole transfer in proteins is, in my opinion, the case of class I ribonucleotide reductase. Here, a network of the type proposed in this paper moves a hole vectorially across 35 Å from one subunit to another and back again to initiate and terminate a turnover <https://www.science.org/doi/full/10.1126/science.aba6794>. The hole is stored on a tyrosine-derived radical or oxidized dimetal cluster ($\text{Mn}^{\text{IV}}/\text{Fe}^{\text{III}}$ or $\text{Mn}^{\text{IV}}/\text{Mn}^{\text{III}}$) in the β subunit and migrates to a cysteine residue in the α subunit via a Trp and 3 Tyr residues [https://www.jbc.org/article/S0021-9258\(21\)00938-8/fulltext](https://www.jbc.org/article/S0021-9258(21)00938-8/fulltext). In the activation of some forms of the β subunit, the Trp residue is oxidized to a transient radical to split the two oxidizing equivalents of a peroxodiiron(III) intermediate, moving one to the surface for quenching (in the manner discussed in this work) and retaining the second internally for production of the tyrosyl radical <https://pubs.acs.org/doi/10.1021/ja001278u>. In the Mn/Fe type, branched pathways involving the same Trp residue but two distinct tyrosines in β mediate hole transfer orthogonally in the activation and catalytic reactions <https://pubs.acs.org/doi/10.1021/bi800881m>. I would think it advantageous to note this enzyme system in lines 41-47.
2. UV-Vis describes an energy range. The spectroscopy is *absorption* spectroscopy. combining them is best (ultraviolet-visible absorption spectroscopy).
3. Line 114: "unprecedented" is hype-y here, and I would delete it. If the results weren't new, they wouldn't be publishable.
4. Many scientists prefer "variant" to "mutant" when they are site-directed and purposefully prepared.
5. Line 171: what is a "subfraction"? Wouldn't "fraction" mean the same thing and be simpler?
6. "Ground state" is a physical chemical term not interchangeable with "resting state," which is what the authors mean in, for example, line 221.

7. I am uncomfortable with the degree of certainty with which the authors attribute some of their absorption features to Tyr or Trp radicals without other evidence. In most cases, I believe the assignments, but there are not, to my knowledge, examples of Tyr radicals with their sharp peak as high in energy as 400 nm, as in, for example, line 230. All that I am aware of fall between ~ 406 and ~ 416 nm. Spectral subtractions, singular value decompositions with resolution of true basis spectra, or EPR spectra could bolster the case for the assignments.

8. In general, the figures are not, in my opinion, of the highest possible quality. For example, Figure 2 would be better with fewer spectra in each panel, possibly distinguished by color, better scaling of x- and y- axes, and some kinetic traces (A-vs.-time) to support the kinetic arguments (e.g., lines 198, 233). The latter traces appear much later, but are implicitly discussed here, and it is frustrating on a first read not to know to look several figures forward. In addition, tyrosyl radical features can often be resolved to very accurate kinetic traces by performing a "drop-line correction," e.g., $A_{412} - (A_{418} + A_{406})/2$. Because these features are so sharp, this procedure can resolve the contribution at the λ_{\max} from the Tyr•.

9. line 339 has N_{π} which must mean N_{δ} ?

10. Lines 355-356 simply restate the prior sentence. The sentence seems redundant.

11. Near line 449, I have the feeling that the authors lose track of what they are describing and conflate a concentration dependence (Figure 4B) with a kinetic measurement. Perhaps the sentence: "In this case, the linear phase of the reaction becomes shorter and initial rates do no longer increase and even go down at increasing H_2O_2 concentration." is not intended to refer to the concentration dependence in Figure 4B, but it is sandwiched between two "calls" to this figure. Clarification might help avoid any misunderstanding.

12. Line 453: "poorer" is an adjective but is used here as an adverb. "Less well" or "more poorly" would be grammatical.

13. Lines 518-519: Why the decimal (as opposed to scientific notation) for the rate constants? At least use commas...

14. Line 526: In a comparison of two things, it is a bit illogical for both to be identical. The "both" is implicit in their identity.

15. I think the discussion is somewhat longer than it needs to be. I appreciate all the authors' points here, but I would prefer to see it limited to the top 3-4 points.

16. I found a few typographical errors in the SI. Here is one line on p. 17, for example: "The perhaps somewhat dependency of the catalytic crate rates on the concentration nof ascorbate is a consequence of the extreme conditions"

Revision of Manuscript NCOMMS-23-44184-T, “Mutational dissection of a hole hopping route that can be tuned to boost peroxygenase activity in LPMOs”, now entitled “Mutational dissection of a hole hopping route that affects the peroxygenase activity and oxidative stability of LPMOs”, by I Ayuso-Fernandez et al.

Detailed description of how the authors have responded to the reviewers’ comments (February 2024)

This document contains all text provided by the reviewers.

Line numbers in our responses refer to the revised manuscript.

Please note that several supplementary figures have been added, as outlined below, and that, therefore, the numbering of several existing supplementary figures has changed.

REVIEWER #1

COMMENT: This is a very important manuscript that demonstrates the presence and functioning of protective hole-hopping pathways in an LPMO enzyme. Presumably, conclusions from this work could be extended and generalized to other oxidases. The authors have identified relevant tryptophan and tyrosine residues in the enzyme structure and established their functions by judicious mutations, replacing Trp and/or Tyr by inactive phenylalanine. It was shown that, in the absence of a substrate, the hole is channeled across an enzyme molecule, starting at a Trp residue close to the Cu center and ending at a surface Tyr/His pair by forming a deprotonated Tyr•. The manuscript makes a plenty of important points about enzyme activity and protection. The very existence of the hopping pathway was demonstrated, mutation of the first Trp residue (5.2 Å from the Cu center) in the pathway affects both protection and enzymatic activity while residues further down the pathway affect only the protection. The middle part of the hopping pathway exhibits some redundancy, whereby the holes can follow different paths toward the same Tyr/His target. It is shown the diminishing the protection activity enhances catalysis but at the expense of enzyme stability – pointing to a competition between substrate oxidation and enzyme protection by hole hopping. The manuscript is very well and clearly written, all results and conclusions are well documented. It is recommended for publication with only a few comments/suggestions:

RESPONSE: We thank the reviewer for the positive evaluation of our work and appreciate the comments and suggestions.

COMMENT: It is recommended to state clearly the nature of the first hopping steps. Is W178 supposed to be oxidized by HO•?

RESPONSE: We agree that defining the first steps would be of great interest, but in the case of LPMOs the catalytic cycle is not fully understood, nor is the nature of the hole that is transferred. We are not in the position to point at a specific species, but we should of course have mentioned the options, which we now have done. One likely scenario involves the HO• radical that is generated after the homolytic cleavage of H₂O₂. Another scenario, which is supported by modelling studies of the emergence of oxidative damage in the catalytic center of LPMOs, entails that the hole would be derived from the copper-oxyl intermediate. We have added a few lines of text (with references, including one that was not yet cited, Hagemann et al., 2024) to describe both scenarios to the introduction (lines 98-102). The added text reads:

“The formation of these radicals is thought to be initiated by highly oxidizing intermediates that emerge during the catalytic cycle and that can initiate amino acid radical formation, namely the

hydroxyl radical formed after homolytic cleavage of H₂O₂ [43], [46], [47] or the subsequently formed copper-oxyl [51], [55].“

COMMENT: W119, presumably the second hopping intermediate, seems to be even less solvent exposed than W178 (Fig. 1 – left). Taking guidance from photolyase, this would imply that W119 oxidation is energetically uphill (which is acceptable for an intermediate hopping step). How would W119•+ be deprotonated? By H160? Is there another proton – accepting residue in the vicinity of the W119 indole NH? Or, at least, a close-lying water molecule? Please comments on these issues.

RESPONSE: We thank the reviewer for drawing our attention to this issue, which made us take a closer look at the environment of W119 and which brought up some new aspects worth mentioning. We have added a new Figure showing structural details, as suggested by this reviewer in a later comment (new Supplementary Figure S15). The Figure shows that there are three crystallographic waters near the W119 indole and, potentially more importantly, the side chain of Asp149. All these could play a role in the deprotonation associated with oxidation of Trp119. Considering the distances, direct proton transfer to His160 seems less likely. However, indirect, water-mediated proton transfer is conceivable, considering the proximity of one of the crystallographic waters to both W119 and H160.

The textual changes made in relation to this comment are described below, in our response to the comment starting with “On page 24, an interesting point is made about the W119-H160 pair.

COMMENT: The discussion of the kinetics of Y121F mutants (p. 13, also p. 24) remains open-ended. What is author's conclusion/preferred mechanism? Could oxidized Y30 and Y39 be readily deprotonated owing to their high water-exposure?

RESPONSE: We agree that this discussion remains open-ended and we have no clear conclusion or preferred mechanism. The key point, addressed implicitly in the Results section and explicitly in the Discussion section, is that there seems to be redundancy in the “downstream” part of the hole hopping pathway. Such redundancy has been observed in other systems. We do not want to speculate more, not in the least because we lack data for W108 variants, while W108 is close to the Y121-H160-W119 cluster (please see our response to the next comment). We think that Y30 and Y39 could be deprotonated easily since they are solvent-exposed. The crystal structure of SmAA10A (2bem) shows water molecules associated to the hydroxyl groups of these two tyrosines. If radical formation at Y30 or Y39 happens remained elusive during our experiments; either such radical formation is too fast to be detected (since the Y121F variant completely lacks a Tyr- signal) or it does not happen at all.

Upon rereading the Results section (p. 13), and with the reviewer’s comment at hand, we did not detect useful textual changes. However, in the Discussion section, when discussing the possible redundancy of dissipation routes, we have made two changes for clarity:

Line 642-643: The words “surface-exposed” have been added to “For example, mutation of surface-exposed Y30 and Y39, led to...”

At the end of this paragraph (lines 646-652), the following sentence has been added after “Still, mutations of Y121 and H160 did not slow down decay of the tryptophanyl signal, which, apparently can dissipate via routes that do not involve the Y121-H160 pair and other, detectable, radical intermediates. This could be routes involving Y30 and Y39, which likely are easily deprotonated since their hydroxyl groups are solvent exposed and show associated water molecules in the crystal structure of the enzyme.”

COMMENT: On p. 24, an interesting point is made about the W119-H160 pair. Histidine oxidation seems thermodynamically impossible. However, a partially delocalized structure involving also the W119 proton could be feasible. In the WT, H160 (which seems to be solvent-exposed) could perhaps play an intermediate role in the electron and proton transfer involving both W119 and Y121. Please comment. A detailed structural picture of this region would help.

RESPONSE: The interactions in the W119-Y121-H160 region are indeed complex and multi-faceted. When looking at the region (new supplementary figure S15, added based on reviewer’s suggestions), all three amino acids are close to each other, whereas, as already alluded to above, W119 and H160 are also close to crystal waters. Our experimental data clearly show that W119 and the H160/Y121 pair are involved in a main hole hopping route. However, when it comes to how exactly holes are

transferred in this region, uncertainties remain and the reviewer is completely right in pointing at possible alternative scenarios, which are reflected in the new Figure S15 and in the revised main text (as detailed below). Of note, Trp108, the role of which we unfortunately could not address by mutagenesis, also comes into play here, as shown in Figure S15. Further studies are needed to fully unravel what is happening and, therefore, the revised (Discussion) text only raises the issues, without excessive speculation. We have rewritten parts of this paragraph, in particular the first and last parts, in response to both this comment and the comment above addressing deprotonation of the W119 radical. The complete revised paragraph reads (lines 664-688, strikethrough text removed here for clarity):

“Supplementary Fig. S15 provides a closer look of the environment of W119 and the Y121-H160 pair. It is worth noting that the indole group of W119 has several crystal waters nearby and is at hydrogen bonding distance of the carboxylic head group of an aspartate (D149). These nearby groups will facilitate proton transfer associated with oxidation of W119. Although it is not possible to pinpoint exactly how holes are removed from the protein after having traversed its core, reaching W119, it is clear that surface-located residues must be essential. The Y121-H160 pair, in which the Nδ of H160 hydrogen bonds (2.7 Å distance) with the hydroxyl of Y121, is of particular interest (see Supplementary Fig. S15). Mutation of these residues in *SmAA10A* led to decreased enzyme redox stability, showing that stability suffers if an escape point for radical transfer is removed. Radical transferring Tyr-O[•]---HN⁺-His pairs have been characterized in other redox systems, including enzymes and photosystem II (PSII) of green plants and cyanobacteria [59], [82]. Considering that Y121 seems responsible for the tyrosyl signal, the effect of the single Y121F mutation on enzyme stability under redox stress is modest, whereas the effect of the H160F mutation is more pronounced. This suggests that hole transfer from core Trp residues to the solvent exposed H160 can occur, bypassing Y121. As shown in Supplementary Fig. S15, H160 is part of a cluster of redox active amino acids (W119 and W108; and, in the wild-type, Y121) that also contains crystal waters close to W119 and H160. Thus, there may be multiple options for hole and associated proton transfer and for the generation of partially delocalized structures that facilitate transfer.”

COMMENT: p. 13, lines 350-355: This sentence is too long a hard to understand. It is recommended to rephrase it and (for comparison) state also the W• fraction detected for the WT (3% from p. 7?).

RESPONSE: The text has been rephrased and now reads:

“The seemingly central role of the Y121/H160 couple in radical dissipation is supported by the strengthened tryptophanyl feature in variants where this couple is disrupted, with the fraction of W• detected increasing from 3% in WT to 12 % in Y121F, 11% in H160F and 20 % Y121F/H160F. In light of this central role, it is somewhat unexpected that the tryptophanyl feature shows similar velocities of formation and decay for WT, Y121F, H160F and Y121F/H160F.”

REVIEWER #2

COMMENT: “Mutational dissection of a hole hopping route that can be tuned to boost peroxygenase activity in LPMOs” by Ayuso-Fernandez et al describes work analysing hole/electron transfer pathways through *SmAA10* (previously known as CBP21) using site-directed mutagenesis. Using a sequence analysis of the AA10 family and a structure guided approach the authors designed mutations in the core Trp and Tyr residues found in this family of enzymes near the enzyme active site to investigate whether they acted as protective hole hopping routes in the protein. Using stopped flow spectrophotometry they were able to observe the appearance of Trp and Tyr radicals in the protein in the absence of substrate and observed differences in the rate at which these radicals formed or decayed as a result of their mutations. Activity assays were also used to investigate the effects of the mutations on substrate activity and on peroxide sensitivity revealing some differences in the behaviour of the mutants compared to the wild type enzyme. Ultimately, the WT enzyme was the most effective enzyme but some useful insights into the potential importance of the core Trp/Tyr

residues was revealed in allowing reactive species to be channelled away to the protein surface during unproductive reactions.

Overall, the paper is very well written and clear, the experiments have clearly been performed to a high standard and the data are well presented. For me, it would be nice to include the WT data for the stopped flow experiments in figure 2 as flicking back several pages to figure 1 for comparison was slightly unsatisfactory, particularly as we tend to read on screens these days.

RESPONSE: We thank the reviewer for the positive evaluation of our work. Following the reviewer's suggestion, WT data is now included in Figs. 2 and 3. Please note that both figures have additional changes (addition of kinetic traces), which were made in response to comment #8 by reviewer #3.

COMMENT: Other than that minor suggestion, there were two main things that stood out to me that should be considered by the authors:

1. To me, the “boosting” of the peroxygenase activity in the W119F mutant is not that exciting. The increase in rate doesn't seem huge and the follow up experiments show marginal differences compared to wild type. It does not result in an overall increase in product yield compared to WT and as the authors admit the enzyme becomes more rapidly inactivated so ultimately the WT enzyme is still the best one here. Whilst the observations with this mutant are interesting, I don't think that there is that significant a “boost” in the peroxygenase activity of the enzyme to suggest that it's going to be a really useful route towards engineering better enzymes for the future. I would therefore suggest putting a little less emphasis on this observation and consider changing the title slightly to something along the lines of “Mutational dissection of a hole hopping route through a bacterial LPMO.” The results are still very interesting and there is no real need to “oversell” this.

RESPONSE: We do not fully agree with the reviewer, since we believe that the observed effects are very interesting. We do agree, however, that we are overselling. The effects are indeed small, but they are clear and significant and show an example of the anticipated but rarely observed trade-off between protection and activity. In this respect, Figure 5 is, in our view, pretty convincing and exciting. It is true that the W119F variant is more rapidly inactivated than the WT, but this is the case when using intentionally damaging reaction conditions (e.g. 300 μM H_2O_2 added in a single dose at the beginning of the reaction; Fig. 4A). When using more controlled low dosing of H_2O_2 (Fig. 4C) the mutant looks better. Still, the effects are small and there is indeed no basis to claim that the mutant generally “performs better” than WT, for example in an applied setting. Further experimentation would be needed to potentially justify such claims. Acknowledging the reviewer's justified critique, we have made the following changes:

- The title of the paper has been changed from “Mutational dissection of a hole hopping route that can be tuned to boost peroxygenase activity in LPMOs” to “Mutational dissection of a hole hopping route that affects the peroxygenase activity and oxidative stability of LPMOs”.
- We have slightly modified the last sentence of the Introduction. We think it is reasonable to keep referring to engineering in general terms (e.g., including introduction of protective measures). We do not claim or suggest anywhere that enzymes can be made better by “blocking” hole hopping routes. The revised sentence no longer contains the word “unprecedented” in front of “insight” (see comment #3 by reviewer #3) and contains two extra words (“understanding and”). It now reads: “Our findings provide insight into hole hopping and LPMO functionality, and open new avenues for understanding and engineering the performance of oxidoreductases.”

COMMENT: 2. My other concern is a technical one that relates to the consequences of introducing mutations into the hydrophobic core of a protein. The authors do not appear to have done anything to investigate the potential effects this could have on the fold of the protein more generally. I appreciate that control experiments using EPR suggest that there is little effect on the active site of the enzyme and the peroxygenase stability measurements show that many of the mutants are more sensitive to oxidative damage as one might expect. My question is, could this also result from a general destabilisation of the protein which could also limit the utility of the hole hopping pathways? The mutations may not necessarily simply block the hole hopping route and so some verification that the overall fold of the protein is unaffected would be a nice addition. A detailed structural analysis probing the protein dynamics in these mutants will undoubtedly be in the future plans of the authors and I am not suggesting that it's necessary to do that here. Some simple thermostability measurements

could be made with the mutants to show that the mutations are not generally destabilising and that would just help to close off that question in terms of introducing mutations into the core of the protein. Addressing this second point will just firm up the conclusions a little bit in what is a very good paper. I very much enjoyed reading the paper and I hope that the authors find these comments useful moving forward.

RESPONSE: First of all, thanks again for constructive comments. This suggestion has led to increased robustness of the paper.

The reviewer is completely right, and we appreciate that the reviewer acknowledges that the provided data so far indicate that the protein variants are fine. We also agree that future studies of core dynamics in the protein variants are of major interest. Following the suggestion of the reviewer, we have investigated the effect of the mutations on the stability of the different variants. We used the SyproOrange thermal shift assay to determine the apparent T_m of all the protein variants and to check for possible destabilizing effects. A new supplementary figure (Supplementary Figure S8) has been added and the experimental method has been added as a supplementary method. The corresponding results are described in lines 398-406. Briefly, all variants show wild-type like unfolding behaviour (i.e., similar curve shape), confirming the results for EPR indicating that the proteins are correctly folded and stable. Looking at the apparent T_m values, almost all the mutations have a destabilizing effect, as one would expect for this type of mutations. However, importantly, the apparent T_m of all the variants is well above the highest T used in the study. Remarkably, one of the most interesting variants (W119F), which receives considerable attention in the manuscript, only shows a minimal change in apparent T_m relative to the WT. Taken together with the data that was already included in the manuscript (e.g., EPR), these thermal stability data confirm that the effects observed are due to hole hopping being manipulated.

The new text at lines 398-406 reads:

“Given that most of the mutations in *SmAA10A* have a negative effect on packing in the protein core, we measured the apparent melting temperature, T_m , to assess whether observed mutational effects could in part be due to destabilization of the protein fold. As expected, most variants show reduced stability relative to the WT enzyme (Supplementary Fig. S8). However, even the lowest T_m observed (56 °C for W178Y/W119F, compared to 71 °C for WT) is well above the maximum reaction temperature used in this study (37 °C), indicating that in every condition used the LPMO variants were well folded and stable. Of note, the W119F variant, which is discussed extensively below, was almost as stable ($T_m = 70$ °C) as the WT enzyme.”

REVIEWER #3

COMMENT: This manuscript by Ayuso-Fernandez and co-workers characterizes variants of a copper-dependent lytic polysaccharide monooxygenase/ peroxygenase from the AA10 family to understand the possible functional significance of tyrosine and tryptophan residues in controlling the potent chemistry leading to carbohydrate oxidative degradation. The thesis is essentially the idea espoused by Gray and Winkler that enzymes that make such potentially oxidizing intermediates have built in fail-safe mechanisms – specifically networks of aromatic residues to pump oxidizing equivalents or "holes" to the protein surface where they can be quenched by any reductant – to make them more robust catalysts. The work uses analytical chemistry, kinetic data, and some spectroscopy to identify a primary hopping pathway that, somewhat surprisingly, seems to trade a modest diminution in efficiency for a substantial gain in robustness. I think the work has value and is generally well done but can be improved with some revisions.

My recommendations:

1. The most fascinating case of functional hole transfer in proteins is, in my opinion, the case of class I ribonucleotide reductase. Here, a network of the type proposed in this paper moves a hole vectorially across 35 Å from one subunit to another and back again to initiate and terminate a turnover <https://www.science.org/doi/full/10.1126/science.aba6794>. The hole is stored on a tyrosine-derived radical or oxidized dimetal cluster (MnIV/FeIII or MnIV/MnIII) in the β subunit and migrates to a cysteine residue in the α subunit via a Trp and 3 Tyr residues [https://www.jbc.org/article/S0021-9258\(21\)00938-8/fulltext](https://www.jbc.org/article/S0021-9258(21)00938-8/fulltext). In the activation of some forms of the β subunit, the Trp residue is oxidized

to a transient radical to split the two oxidizing equivalents of a peroxodiiron(III) intermediate, moving one to the surface for quenching (in the manner discussed in this work) and retaining the second internally for production of the tyrosyl radical <https://pubs.acs.org/doi/10.1021/ja001278u>. In the Mn/Fe type, branched pathways involving the same Trp residue but two distinct tyrosines in β mediate hole transfer orthogonally in the activation and catalytic reactions <https://pubs.acs.org/doi/10.1021/bi800881m>. I would think it advantageous to note this enzyme system in lines 41-47.

RESPONSE: We thank the reviewer for this highly justified suggestion. We have now added a sentence pointing at Class I ribonucleotide reductases that includes four new references. The added text reads (lines 46-49):

“Hole hopping is not limited to monomeric proteins, as illustrated by class I ribonucleotide reductases, where catalysis involves a hole traversing from one subunit to another through a chain of aromatic amino acids covering an impressive distance of 35 Å [8]–[11].”

COMMENT: 2. UV-Vis describes an energy range. The spectroscopy is *absorption* spectroscopy. combining them is best (ultraviolet-visible absorption spectroscopy).

RESPONSE: Our apologies for unprecise phrasing. We have made the necessary changes throughout the manuscript and the SI.

COMMENT: 3. Line 114: "unprecedented" is hype-y here, and I would delete it. If the results weren't new, they wouldn't be publishable.

RESPONSE: Good point. The word “unprecedented” has been deleted from “Our findings provide unprecedented insight into....”

COMMENT: 4. Many scientists prefer "variant" to "mutant" when they are site-directed and purposefully prepared.

RESPONSE: We have made changes throughout the manuscript and the SI, replacing “mutant” by “variant” in most cases.

COMMENT: 5. Line 171: what is a "subfraction"? Wouldn't "fraction" mean the same thing and be simpler?

RESPONSE: We have replaced “subfraction” by “fraction”.

COMMENT: 6. "Ground state" is a physical chemical term not interchangeable with "resting state," which is what the authors mean in, for example, line 221.

RESPONSE: We thank the reviewer for pointing this out and apologize for the mistake. “Ground state” has been replaced by “resting state” throughout (twice in the SI, three times in the manuscript).

COMMENT: 7. I am uncomfortable with the degree of certainty with which the authors attribute some of their absorption features to Tyr or Trp radicals without other evidence. In most cases, I believe the assignments, but there are not, to my knowledge, examples of Tyr radicals with their sharp peak as high in energy as 400 nm, as in, for example, line 230. All that I am aware of fall between ~ 406 and ~ 416 nm. Spectral subtractions, singular value decompositions with resolution of true basis spectra, or EPR spectra could bolster the case for the assignments.

RESPONSE: We thank the reviewer for pointing at the deviation between typical spectral features of tyrosine radicals and our data. We fully agree with the reviewer. Indeed, we should have paid more attention to this issue, although we, based on our mutagenesis work, believe that our assignments are correct (the reviewer seems to agree to this). Our assignments of the Tyr• and Trp• radical absorption features are made primarily based on the mutagenesis work. The absence of any radical signal in the W178F or W178M variants is a good indication that the 1st position of the hole hopping route is W178. Replacing the same 178 position for a Tyr led to a stronger feature at 400 nm, while the assigned Trp• radical feature disappears. The double mutant W178Y/W119F reveals a similar Tyr• radical feature, again with no Trp• signal. We believe that this mutagenesis work presents strong evidence that the broad feature at 505 nm represents a Trp• radical, well in line with literature data. The data also support that the 400 nm feature observed (only) in variants containing the W178Y

mutation reflects a Tyr• radical. The fact remains that the wavelength, 400 nm, is unusual. Another fact is that, while the blue shift and the change in the curve shape are real, the deeper analysis of the data suggested by the reviewer (spectral subtractions) shows that it is not really a shift to a 400 nm feature, but rather peak broadening and that the signal is quite compatible with usual Tyr• radical signals.

We have followed the reviewer's suggestion of doing spectral subtractions and we have included these data for WT and the W178 variants in the Supplementary Information (new Fig. S5). The subtractions show that the feature assigned to a Trp• radical (visible in the WT spectra) indeed is broad (505 – 530 nm), as is commonly observed for such radicals in proteins. The spectral subtractions shed more light on the spectral character of the Tyr• radical. Interestingly, a lower energy shoulder at approximately 400 nm is observed in the WT difference spectra, confirming its presence, also in the wild-type. We note that pure tyrosine radicals show a split absorption spectrum with a second feature at approximately 15 nm less than its maximum at 410 nm (Bansal, K. & Fessenden, R., *Pulse Radiolysis Studies of the Oxidation of Phenols by SO₄⁻ and Br₂⁻ in Aqueous Solutions*, 1976, *Radiation Research*, 67, 1-8). The spectral subtraction of W178Y variants confirms the blue shift but shows that this effect is more of a peak broadening effect (in the 400-415 nm region), rather than a clear shift in the peak from approximately 412 nm to 400 nm. This supports the assignment as a Tyr• radical signals. It is unknown how the protein environment perturbs the electronics of the radical center. The mutational data indicate that the broader feature in W178Y containing variants reflects the Tyr• radical; it seems conceivable that the electronics of the introduced Tyr are perturbed relative to the other Tyr residues in the protein. Of note, the Tyr residue generating the radical signal in the WT enzyme is likely not the same as the Tyr residue generating the radical signal in the W178Y variants.

Although the W178Y variants show slow decrease of the putative Tyr• radical, attempts to manually freeze (mixing in a EPR tube and freezing in liquid nitrogen at slow timed points, i.e. 5 seconds) have been successful and did not yield any clear EPR results. We are currently working on developing freeze-quenching and EPR protocols that may allow us to gain deeper insights into amino acid radical formation.

To address the issue, one sentence has been changed and a short paragraph has been added to the manuscript:

Lines 237-238: “the tyrosyl feature was smoother and more clearly defined than in the WT, and blue-shifted from 412 nm to 400 nm, while”
have been changed to:

“the tyrosyl feature was smoother and broader than in the WT, with an apparent blue-shift from 412 nm to 400 nm, while”

Lines 244-255, this text has been added:

“Assignment of the 400 nm feature in the spectra of the W178Y variants to a tyrosyl radical is not obvious, since the energy of this radical is high, compared to tyrosyl radicals in solution [61], or in natural [43], [62] and designed proteins [2]. Spectral subtractions to remove background signals (Supplementary Fig. S5) confirmed that a blue shift indeed occurs when comparing the WT enzyme with W178Y containing variants but that this is more of a peak broadening effect rather than a true shift from 412 nm to 400 nm. The difference spectra suggest that the 412 nm feature is still present in the W178Y containing variants and also show a weak 400 nm feature in WT spectra. This higher energy feature could correspond to a second feature observed usually at 15 nm less than the maximum signal for tyrosyl radicals in solution [63]. It is conceivable that the observed blue shift is due to the (non-natural) environment of the tyrosine in position 178.”

The spectral subtractions also showed that there is a minute Tyr feature visible for the W178F and W178M mutants. This is now acknowledged in the paper (lines 234-245) where

“Mutation of W178 showed that W178F and W178M are UV-Vis silent (Fig. 2, bottom right panels), whereas W178Y.....”

has been changed to

“Mutation of W178 showed that W178F and W178M are UV-Vis silent (Fig. 2, bottom right panels), although a minute putative Y' feature became visible upon spectral subtraction (Supplementary Figure S5). On the other hand, W178Y.....”.

COMMENT: 8. In general, the figures are not, in my opinion, of the highest possible quality. For example, Figure 2 would be better with fewer spectra in each panel, possibly distinguished by color, better scaling of x- and y- axes, and some kinetic traces (A-vs.-time) to support the kinetic arguments (e.g., lines 198, 233). The latter traces appear much later, but are implicitly discussed here, and it is frustrating on a first read not to know to look several figures forward. In addition, tyrosyl radical features can often be resolved to very accurate kinetic traces by performing a "drop-line correction," e.g., $A_{412} - (A_{418} + A_{406})/2$. Because these features are so sharp, this procedure can resolve the contribution at the λ_{\max} from the Tyr•.

RESPONSE: We thank the reviewer for the suggestions for improving the quality of the figures. Figs 2&3 now include kinetic traces for the assigned radicals, and the axes have been adjusted. The revised figures have more labeled at x-axes, and the y-axes have been adjusted so that they are identical for the formation and decay phases. We have kept the summary figure with all the kinetic traces (Figure 6) at the end of the paper, because we believe it provides a useful overview that supports the Discussion. The number of traces or the colors have not been changed. We believe that the change in shading is sufficient to follow the evolution of the traces and that including more colors would make the figure more "busy" and complex.

We appreciate the suggestion to perform the "drop-line correction" for quantification of the tyrosyl radical. We performed quantification of radicals based on a method used in recent literature for quantifying radicals detected in LPMOs (Jones et al 2020 PNAS; Zhang et al 2023 JACS) and we use quantification only for comparison of the variants that we study. We believe that it is useful for the field that we stick to our original method.

COMMENT: 9. line 339 has $N\pi$ which must mean $N\sigma$?

RESPONSE: Sorry for the mistake. This should have been $N\delta$ and this has now been corrected.

COMMENT: 10. Lines 355-356 simply restate the prior sentence. The sentence seems redundant.

RESPONSE: Although it seems redundant, we prefer to keep this short sentence as it may improve clarity for non-expert readers. Of note, the preceding, complicated sentence has been rephrased in response to the last comment of Reviewer #1.

COMMENT: 11. Near line 449, I have the feeling that the authors lose track of what they are describing and conflate a concentration dependence (Figure 4B) with a kinetic measurement. Perhaps the sentence: "In this case, the linear phase of the reaction becomes shorter and initial rates do no longer increase and even go down at increasing H₂O₂ concentration." Is not intended to refer to the concentration dependence in Figure 4B, but it is sandwiched between two "calls" to this figure. Clarification might help avoid any misunderstanding.

RESPONSE: We thank the reviewer for bringing this confusing text fragment to our attention. Figure 4B shows the results of individual kinetic measurements for each enzyme variant at multiple H₂O₂ concentrations. The highlighted sentence, "In this case...", refers to the underlying data, not to Figure 4B specifically. That is confusing. We have now rephrased the text and hope that things are clearer. The revised text reads (lines 467-472):

"As expected, the rate of the peroxidase reaction, $\Delta\text{Abs}/\text{min}$, increases with the H₂O₂ concentration, and this increase stops when H₂O₂ concentrations become damaging (Fig. 4 B). At higher H₂O₂ concentrations enzyme activation is expected to occur increasingly early and this was reflected in a shortening of the linear portions of the product formation curves (not shown) that were used to calculate the rates. Fig. 4B shows that...."

COMMENT: 12. Line 453: "poorer" is an adjective but is used here as an adverb. "Less well" or "more poorly" would be grammatical.

RESPONSE: We have replaced "poorer" by "less well".

COMMENT: 13. Lines 518-519: Why the decimal (as opposed to scientific notation) for the rate constants? At least use commas...

RESPONSE: Our apologies; this was a mistake that now has been corrected. Scientific notation is now used (as was used in Figure 5 and Supp. Fig S11 in the original manuscript).

COMMENT: 14. Line 526: In a comparison of two things, it is a bit illogical for both to be identical. The “both” is implicit in their identity.

RESPONSE: We have rephrased the text, which now reads: “The two enzymes give similar spectra (Fig. 4D),.....”

COMMENT: 15. I think the discussion is somewhat longer than it needs to be. I appreciate all the authors' points here, but I would prefer to see it limited to the top 3-4 points.

RESPONSE: We have carefully considered this comment and concluded that we would like to keep the Discussion approximately as is. We believe that, with one exception (see below), each point that we address is worth addressing and the reviewer does not seem to disagree. We note that the other two reviewers do not suggest shortening and that both explicitly state that the manuscript was well written and clear.

We have deleted one paragraph, which in fact was quite redundant. The reference to Fig. 6 that appeared in this paragraph has been moved to the first paragraph of the Discussion (which is more appropriate), whereas reference 79 (Zhao et al., 2023, JACS) is now cited in the preceding paragraph. The deleted paragraph:

“At the time of preparation of this manuscript, a study focusing on radical formation for the axial Y in AA9 LPMOs pointed at the role of this Tyr in protecting the copper-binding histidines [81], but little is known about further hole hopping path(s) and hole hopping-related diversity in AA9s. Here we show that AA10s display a different mechanism for protection against oxidative damage, in which a conserved Trp facilitates hole dissipation by connecting the catalytic copper and its coordinating histidines to hole hopping paths that traverse the enzyme (Fig. 6).”

We have also deleted a sentence at the end of a paragraph further down (lines 652-654) because it was redundant and speculated on an aspect which reviewer #2 asked us to tone down a bit. The deleted sentence:

“It is worth noting that the “downstream” mutations did not lead to increases in the initial catalytic rate, contributing to the notion that none of these mutations stopped holes from dissipating away from W178 and the copper site.”

The paragraph starting with “Although it is not possible.....” has been partly rewritten (in response to a comment by reviewer #1) and shortened.

COMMENT: 16. I found a few typographical errors in the SI. Here is one line on p. 17, for example: "The perhaps somewhat dependency of the catalytic crate rates on the concentration nof ascorbate is a consequence of the extreme conditions"

RESPONSE: We have reviewed the SI and corrected typographical errors, including the one pointed at by the reviewer.

ADDITIONAL CHANGES:

- In line 37, we have replaced “after the great oxidation event” by “during evolution”, which is more correct.
- Line 386: “are” added (correction of grammar).
- Past tense has been replaced by present tense in a few instances.
- Line 843: “kinetics” -> “spectroscopy”
- Figure 2 and Figure 3: In the structural picture in the upper left corner, distances between W119 and H160, W119 and W108 and Y121 and W108 have been adjusted and are now indicated by using the same atom pairs as those used in new Fig. S15 (i.e., the closest distances). This was done to avoid confusión and inconsistency but has no significant effect on what Figs 2 and 3 show.

REVIEWERS' COMMENTS

Reviewer #1 (Remarks to the Author):

I am fully satisfied with author's response to my previous comments. My previous overall evaluation still holds; and the manuscript has been improved further. It's ready to be published.

Response to Reviewer #3

Reviewer #3 made several mostly formal comments, which were all addressed satisfactorily. Several other comments required a more detailed response:

Comment 1: The ribonuclease reductase indeed is a fascinating case of hole hopping through aromatic amino acid chains but its purpose is different from the case studied herein: functional vs. protective. In the revision, the authors have mentioned this enzyme in the Introduction and added references, which is perfectly sufficient.

Comment 7 deals with assigning the radical spectra, namely Tyr. This is not an easy question since radical spectra often strongly depend on the molecular environment. The authors convincingly argue based on spectral effects of the mutations. Following the reviewer's suggestion, they also performed spectra subtraction that revealed more details about the character of the shifted Tyr-radical band. Author's response is satisfactory but small changes (that can be made in the proofs) are needed in the revised text:

"...since the energy of this radical is high..." It's not the energy of the radical but energy of its electronic absorption band!

The sentence: "This higher energy feature could correspond to a second feature observed usually at 15 nm less than the maximum signal for tyrosyl radicals in solution [63]." is understandable to the reviewer but not very clear. It should be reformulated. It could be easier to discuss the spectra just in the terms of wavelengths.

Comment 8 on the figures was addressed satisfactorily.

Comment 15 on the Discussion: The authors made some clarifications and removed one redundant paragraph. The Discussion reads well and no further reduction is needed.

All other comments were addressed satisfactorily.

Reviewer #2 (Remarks to the Author):

The authors have addressed all of my questions.

Detailed description of how the authors have responded to the reviewers' comments (April 2024)

Reviewers #1 and #2 were fully satisfied with the revision. There were, however, some minor comments of our responses to reviewer #3. The overview below contains all text provided by this reviewer (labeled with "COMMENT").

COMMENT: Reviewer #3 made several mostly formal comments, which were all addressed satisfactorily.

Several other comments required a more detailed response:

Comment 1: The ribonuclease reductase indeed is a fascinating case of hole hopping through aromatic amino acid chains but its purpose is different from the case studied herein: functional vs. protective. In the revision, the authors have mentioned this enzyme in the Introduction and added references, which is perfectly sufficient.

RESPONSE: we appreciate the insight of the reviewer and we agree with the comment.

COMMENT: Comment 7 deals with assigning the radical spectra, namely Tyr. This is not an easy question since radical spectra often strongly depend on the molecular environment. The authors convincingly argue based on spectral effects of the mutations. Following the reviewer's suggestion, they also performed spectra subtraction that revealed more details about the character of the shifted Tyr-radical band. Author's response is satisfactory but small changes (that can be made in the proofs) are needed in the revised text:

"...since the energy of this radical is high..." It's not the energy of the radical but energy of its electronic absorption band!

The sentence: "This higher energy feature could correspond to a second feature observed usually at 15 nm less than the maximum signal for tyrosyl radicals in solution [63]." is understandable to the reviewer but not very clear. It should be reformulated. It could be easier to discuss the spectra just in the terms of wavelengths.

RESPONSE: We thank the reviewer for this insightful comment and the request for clarification between the energy of the radical or its electronic absorption band, and the use of wavelength versus energy. We have corrected the mistake and, following the reviewer's suggestions, we have simplified the discussion referring to just wavelengths.

COMMENT: Comment 8 on the figures was addressed satisfactorily.

RESPONSE: Thank you

COMMENT: Comment 15 on the Discussion: The authors made some clarifications and removed one redundant paragraph. The Discussion reads well and no further reduction is needed.

RESPONSE: Thank you.

COMMENT: All other comments were addressed satisfactorily.

RESPONSE: We appreciate the final positive comments of this reviewer.